# Sloth: scaling laws for LLM skills to predict multi-benchmark performance across families

**Felipe Maia Polo**[1], **Seamus Somerstep**[1]
**Leshem Choshen**[2,3]   **Yuekai Sun**[1]   **Mikhail Yurochkin**[4]
[1]Department of Statistics, University of Michigan
[2] MIT-IBM Watson AI Lab, IBM Research
[3] Computer Science and Artificial Intelligence Laboratory, MIT
[4] Institute of Foundation Models, MBZUAI

## Abstract

Scaling laws for large language models (LLMs) predict model performance based on parameters like size and training data. However, differences in training configurations and data processing across model families lead to significant variations in benchmark performance, making it difficult for a single scaling law to generalize across all LLMs. On the other hand, training family-specific scaling laws requires training models of varying sizes for every family. In this work, we propose Skills Scaling Laws (SSLaws, pronounced as `Sloth`), a novel scaling law that leverages publicly available benchmark data and assumes LLM performance is driven by low-dimensional latent skills, such as reasoning and instruction following. These latent skills are influenced by computational resources like model size and training tokens, but with varying efficiencies across model families. `Sloth` exploits correlations across benchmarks to provide more accurate and interpretable predictions while alleviating the need to train multiple LLMs per family. We present both theoretical results on parameter identification and empirical evaluations on 12 prominent benchmarks, from Open LLM Leaderboard v1/v2, demonstrating that `Sloth` predicts LLM performance accurately and offers insights into scaling behaviors for complex downstream tasks, increased test-time compute, and compute-optimal scaling of skills. Our code can be found on https://github.com/felipemaiapolo/sloth.

## 1   Introduction

Large Language Model (LLM) scaling laws for benchmarks and downstream tasks efficiently predict the performance of an LLM based on its parameter count and training set size. However, variations in training configurations and data processing across different model families often lead to significant differences in benchmark performance, even for models with comparable compute budgets [Ruan et al., 2024]. Consequently, a single scaling law typically fails to predict performance across all LLMs accurately [Choshen et al., 2024]. In contrast, creating family-specific scaling laws requires training multiple models of increasing size, which is resource-intensive.

In this work, we propose a new class of scaling laws called `Sloth` to solve this dilemma. These scaling laws are fitted using publicly available data (*e.g.*, from LLM leaderboards) across multiple benchmarks, leveraging information shared among benchmarks and model families to improve prediction power and interpretability through parameter efficiency, *i.e.*, fewer parameters without hurting performance. Specifically, we utilize the correlations in benchmark scores to simplify the scaling law in terms of parameter count without harming prediction power by assuming that LLM performance is driven by a set of low-dimensional latent skills, such as reasoning and instruction

39th Conference on Neural Information Processing Systems (NeurIPS 2025).

following, which can be easily interpreted. Furthermore, we hypothesize that these latent skills are similarly influenced by computational resources, such as model size and training tokens, across different LLM families, with the key distinction being each family's efficiency in converting compute into skill levels–which can be estimated with only one model per family. In summary, our main contributions are

- Introducing a new class of scaling laws, `Sloth`, that borrows strength across the available benchmarks and LLM families to make more accurate and interpretable performance predictions of (hypothetical) LLMs in given benchmarks of interest. Specifically, we assume that benchmark performances directly depend on low-dimensional LLM skills, which are influenced by factors such as the number of training tokens and the number of parameters.

- Providing a theoretical result regarding the identification of `Sloth`'s parameters and empirically demonstrating that our scaling laws can (i) accurately predict the performance of large models in 12 prominent LLM benchmarks and (ii) provide interpretable insights into LLM skills and scaling behavior.

- Demonstrating how predicted latent skills and our model can be used to (i) predict model performance in complex downstream tasks that involve coding and emotional intelligence, (ii) predict LLM behavior with scaled test-time compute, and (iii) derive optimal-scaling rules for skills.

## 1.1 Related work

**Scaling laws for deep neural networks:** In recent years, researchers have studied scaling laws from different angles. Rosenfeld et al. [2019] provides experimental scaling laws that predict model loss as a function of training set size, model width, and model depth. Likewise, Kaplan et al. [2020] establishes scaling laws that primarily measure loss (perplexity) and not accuracy on downstream tasks or benchmarks. Motivated by the presence of hard limits on the size of trainable data sets but a hypothetical unlimited ability to scale models, the authors of Muennighoff et al. [2023] establish scaling laws in constrained data settings. They find that perhaps unsurprisingly, increasing computing provides diminishing returns if data does not scale. Gadre et al. [2024] addresses the gap between the assumptions in scaling laws and how training is performed in practice; in particular, they construct scaling laws that both perform well in the over-training regime and predict performance on downstream tasks. In a similar but distinct direction, some works try not only to estimate scaling laws but also respond to the following strategic question: "Given a fixed FLOPs budget, how should one trade-off model size and the number of training tokens?" For example, Hoffmann et al. [2022] provides a partial answer, introducing the celebrated family of Chinchilla scaling laws and finding that training tokens and parameter size should roughly scale together. This contrasts with the older work of Kaplan et al. [2020] that provides a series of power laws that imply that simply increasing parameter count will provide good returns. Each of these referenced works trains models with a particular pretraining setting (e.g., architecture) at various sizes and ultimately seeks to predict test loss. Our focus is distinct, we fit scaling laws on existing benchmark data of multiple model families and predict LLM benchmark performance with minimal amount of data on the new family being predicted. Even though Allen-Zhu and Li [2024], study how LLMs can retain knowledge depending on their scale, the closest related works are Owen [2024], Ruan et al. [2024], Gadre et al. [2024]; we will provide a detailed comparison with their work throughout the paper.

**LLMs latent skills:** Given that the performance of large language models (LLMs) in different and diverse benchmarks is correlated, it makes sense to think that those models have some low-dimensional latent skills that are reflected in downstream tasks. In this direction, Ilić [2023] extracts a general intelligence factor ("g-factor") for LLMs using the Open LLM Leaderboard [Beeching et al., 2023] and GLUE [Wang et al., 2018] using factor analysis. They also verify that this "g-factor" positively correlates with model size. In a similar direction Burnell et al. [2023] uses HELM [Liang et al., 2022] data to reveal that LLM intelligence may be constituted by three distinct, yet correlated factors. They also verify a positive correlation between model size and these latent skills, yet they do not propose a formal scaling law. In their study, the authors do not account for the training set size or model family information, leading to a poor fit of the regression model; this leaves good extrapolation as an open problem we address. In Kipnis et al. [2024], a unidimensional item response theory model is applied to each one of the 6 (filtered) benchmarks of the Open LLM Leaderboard. A factor analysis on the skill parameters shows that the main factor (carrying 80% of the data variability) is highly correlated with the "grand" (average) score of LLMs. In a related but different direction, Maia Polo et al. [2024a,b] show that inferring low-dimensional latent skills of LLMs can make model evaluation much more efficient, saving up to 140x in computing power. In this work, we explicitly model LLM

skills as a function of computing resources, which enables the creation of accurate and interpretable scaling laws for benchmark performances.

## 2 Scaling laws for benchmark data

### 2.1 Problem Statement

In this section, we describe the setup we work on and what our objectives are. Within a family of LLMs $i$ (*e.g.*, LLaMA 3), our objective is to estimate the performance of a big LLM, *e.g.*, with 70 billion parameters, in a benchmark $j$, *e.g.*, MMLU, given evaluation data from smaller models in the same family. Let $s$ represent the size of the LLM, defined as the number of parameters, and let $t$ denote the number of training tokens. We define $Y_{ij}(s,t) \in [0,1]$ as the score of an LLM from family $i$, with size $s$ and trained on $t$ tokens, on benchmark $j$. Our goal is to approximate:

$$\mu_{ij}(s,t) = m[Y_{ij}(s,t)]. \tag{2.1}$$

Here, $m[\cdot]$ should be interpreted as a central tendency summary measure of a random variable, such as the mean or median[1]. Ideally, the model for $\mu_{ij}$ will be simple and some of its parameters will be shared among model families and benchmarks; in this case, the model becomes more interpretable and more data can be used in the fitting process, making the model better estimated. From now on, we denote the set of model families as $\mathcal{I} = \{1, \cdots, I\}$ and the set of benchmarks as $\mathcal{J} = \{1, \cdots, J\}$.

### 2.2 Previous approaches to scaling laws for benchmarks

The closest works to ours that propose models for $\mu_{ij}(s,t)$ (2.1) are Owen [2024], Ruan et al. [2024], and Gadre et al. [2024]. While Gadre et al. [2024] indirectly model the quantity of interest via the LLMs perplexity in specific datasets, which might not be readily available, Owen [2024] and Ruan et al. [2024] model $\mu_{ij}(s,t)$ directly through a regression model connecting compute and benchmark performance. One assumption they made is that the performance on benchmarks only depends on $s$ and $t$ through the total amount of training FLOPs, which can be approximated by $c(s,t) = 6st$. That is, if $\sigma : \mathbf{R} \to [0,1]$ denotes a fixed activation function, *e.g.*, the standard logistic (sigmoid) function, and $\gamma_j \in [0,1]$, then it is assumed that

$$\mu_{ij}(s,t) = \gamma_j + (1 - \gamma_j)\sigma(\eta_{ij}(s,t)), \tag{2.2}$$

where $\eta_{ij} : \mathbf{R}^2 \to \mathbf{R}$ denotes a linear predictor such that $\eta_{ij}(s,t) = \alpha_{ij} + \beta_{ij} \log c(s,t)$, which can be easily interpreted. Here, $\gamma_j$ adjusts the lower asymptote of $\mu_{ij}$ and accounts for the probability of LLMs scoring correctly by chance. Owen [2024], in their best performing models, considers the case in which $\gamma_j = 0$ (or adds a similar offset parameter to the model) and the parameters $\alpha_{ij}$ and $\beta_{ij}$ are independent of the model family $i$. On the other hand, Ruan et al. [2024] consider both $\alpha_{ij}$ and $\beta_{ij}$ to be family-dependent and, in their most general model, $\gamma_j$ can assume values in $[0,1]$.

The biggest issue with previous approaches when modeling $\mu_{ij}$ is that they are either too restrictive or too flexible. From the restrictive side, they assume that (i) $\mu_{ij}$ depends on $s$ and $t$ only through FLOPs, (ii) there are no family-dependent parameters, or (iii) the activation function $\sigma$ is fixed and well-specified. From the flexibility side, Ruan et al. [2024] assume both $\alpha_{ij}$ and $\beta_{ij}$ to be family dependent making estimation hard (or impossible) depending on the number of models we see for each family. From Ruan et al. [2024]: "(...) fitting such a scaling law can be tricky, as each model family $f$ and downstream benchmark has its own scaling coefficients $\beta_f$ and $\alpha_f$. This means that scaling experiments, especially for post-training analysis, are often fitted on very few (3-5) models sharing the same model family (...)." Thus, in their experiments, they consider a different problem setting, where a large LLM has been trained and evaluated on some benchmarks and use their method to predict its performance on other benchmarks.

At the end of the day, Owen [2024] and Ruan et al. [2024] end up working in different setups: Owen [2024] does not use family information at prediction time, making their scaling law less accurate but more generalizable, and Ruan et al. [2024] assume families are important at prediction time but consider that the target model has already been trained, making their scaling law less applicable in practice and more interesting from an interpretability point of view. In this work we wish to instead predict the performance of a larger LLM *without having to train it* but taking family information into account, thus allowing practitioners to make decisions regarding investing resources into training

---

[1]Because we minimize the Huber loss in this paper, we aim to approximate the median

large LLMs. Moreover, our formulation also allows interpretable insights from the data. Despite different setups, we make comparisons with Owen [2024] and Ruan et al. [2024] throughout this work by considering their applications/adaptations as baselines.

## 3   Scaling laws for LLMs skills with `Sloth`

### 3.1   Model architecture

We present a novel scaling law called `Sloth`, which introduces several modifications to (2.2). The key innovation of `Sloth` lies in its explicit modeling of the correlation structure between benchmarks, resulting in improved predictive accuracy and interpretability. Moreover, `Sloth` proposes that (i) LLM capabilities should scale with computing resources similarly across families up to an efficiency factor, (ii) benchmark performance can depend on $s$ and $t$ not only through the total number of FLOPs, and (iii) that the function $\sigma$ can also be learned in cases in which predictive performance is important. We detail these points.

Inspired by the latent skills (*e.g.*, reasoning, language modeling, instruction following) inferred from benchmark data in Burnell et al. [2023], Ilić [2023], Ruan et al. [2024], Gor et al., Kipnis et al. [2024], Maia Polo et al. [2024a,b], we propose creating a scaling law for LLMs skills by leveraging the correlation structure of the benchmarks; for example, we model how the construct "reasoning" scales with compute instead of modeling benchmarks scores directly. The two major advantages of this approach are better performance prediction since we have fewer parameters to fit (reducing overfitting) and extra interpretability/insights. Concretely, we model $\eta_{ij}(s,t)$'s simultaneously for benchmarks $j \in \mathcal{J}$ as each being a linear combination of the same low-dimensional latent skills $\theta_i(s,t) \in \mathbf{R}^d$ plus a bias term $b \in \mathbf{R}^J$, where $d \ll J = |\mathcal{J}|$. Denote $\eta_i(s,t) \in \mathbf{R}^J$ as the vector of $\{\eta_{ij}(s,t)\}_{j \in \mathcal{J}}$. Mathematically, we have

$$\eta_i(s,t) = \Lambda \theta_i(s,t) + b. \tag{3.1}$$

One can see that $\Lambda \in \mathbf{R}^{J \times d}$ encodes the correlation structure between the benchmarks; in particular, it tells us which benchmark measures overlapping (or distinct) skills. Interestingly, our model has a strong connection with factor analysis (FA) models, which we elaborate on in detail in Appendix C. In FA, the matrix $\Lambda$ is known as factor loadings while $\theta_i(s,t)$ are known as factors.

Next, we propose a model for $\theta_i(s,t)$. Inspired by models used in Economics, we use the family of translog production functions from stochastic frontier analysis [Kumbhakar and Lovell, 2003]:

$$\theta_{ik}(s,t) = \alpha_{ik} + \beta_k^\top x(s,t); \quad 1 \le k \le d, \tag{3.2}$$
$$x(s,t) = (\log(s), \log(t), \log(s)\log(t)).$$

Note that (i) the intercept parameter $\alpha_{ik}$ is indeed family-dependent while each skill slope is shared across families and (ii) $\theta_i$ can depend on $s$ and $t$ not only through $c(s,t)$. In economic terms, the intercept term $\alpha_{ik}$ can be interpreted as an efficiency measure of the family $i$ in converting compute to performance for skill $k$ and, in practice, will absorb all hidden factors specific to family $i$ such as data quality, post-training factors, *etc.*. We note that the interaction term in $(\log(s)\log(t))$ accounts for the fact that the impact of $\log(s)$ and $\log(t)$ on skills might depend on each other; in Appendix D, we show some evidence that this is indeed the case. Additionally to the changes in $\eta_{ij}$, we propose making the activation function $\sigma$ trainable and specific to each benchmark $j$ if needed. To that end, we adopt a semi-parametric single-index model using neural networks [Bietti et al., 2022]. To make the results more behaved if (out-of-support) generalization is needed, we assume $\sigma_j : \mathbf{R} \to [0,1]$ is given by a monotonic (increasing) neural network, which can be achieved by constraining its weights to be non-negative [Sill, 1997] and its last activation function to be sigmoid. We note, however, that one can always forgo training of the link function and instead assume a sigmoid structure as this simplifies model fitting and may make the model more interpretable. We give more details about model fitting in Section 3.2. Since `Sloth` is a simple neural network, both model fitting and prediction are done within seconds by a commercial laptop.

## 3.2 Model fitting

Assume that for each model family $i$ we observe a set of tuples $(s,t)$'s denominated by $\mathcal{E}_i$. Then, we fit the model by solving the following minimization problem

$$(\hat{\gamma}, \hat{\sigma}, \hat{b}, \hat{\Lambda}, \hat{\alpha}, \hat{\beta}) = \underset{\substack{\gamma_j \in [0,1],\, \text{for } j \in \mathcal{J} \\ \sigma_j : \mathbf{R} \to [0,1] \text{ increasing },\, \text{for } j \in \mathcal{J} \\ b_j \in \mathbf{R},\, \text{for } j \in \mathcal{J};\, \Lambda \in \mathbf{R}^{J \times d} \\ \alpha_{ik} \in \mathbf{R},\, \text{for } i \in \mathcal{I} \text{ and } 1 \leq k \leq d \\ \beta_k \in \mathbf{R}^3,\, \text{for } 1 \leq k \leq d}}{\arg\min} \sum_{i \in \mathcal{I}} \sum_{(s,t) \in \mathcal{E}_i} \sum_{j \in \mathcal{J}} \ell_\delta(\mu_{ij}(s,t), Y_{ij})$$

where $\ell_\delta$ is given by the Huber loss with hyperparameter $\delta = .01$ and $\mu_{ij}(s,t)$ denotes the most general version of our model. We minimize the loss function via gradient descent using the Adam optimizer [Kingma and Ba, 2017] with a decaying learning rate. We parameterize $\gamma_j$ using the sigmoid transformation to guarantee the constraints are satisfied. Similarly, we truncate the weights of the two-hidden-layer neural network $\sigma_j$ to ensure the trainable function is increasing. If one desires, $\sigma_j$'s can be set to fixed functions, *e.g.*, sigmoid, and $\gamma_j$'s can be fixed beforehand. Unfortunately, the minimization problem is not convex as expected when fitting factor-analysis-like models; multiple initializations of the optimizer can be applied to guarantee a better fit.

## 3.3 Interpretability and practical considerations post model fitting

Ideally, to make reasonable interpretations of models like `Sloth`, we need its parameters to be identifiable, *i.e.*, the map connecting $\eta_{ij}(s,t)$'s and model parameters should be bijective. Unfortunately, as in all exploratory factor models, this is not the case. However, we theoretically show in Appendix A that the model parameters are identifiable up to some parameter transformations. This observation allows us to find a configuration that makes the model more interpretable by mirroring a standard approach used in factor analysis, *e.g.*, in Chen et al. [2019]'s applications. The main idea is that we fit `Sloth` without any constraints and then find a configuration of skills (using factor rotation) that makes results interpretable. We detail the applied process in Appendix A.1.

## 4 `Sloth` in practice

In this section, we present experimental results that provide evidence of the usefulness of `Sloth`. We perform experiments on a set of twelve benchmarks and state-of-the-art LLM families, including LLaMa 3 [Dubey et al., 2024], Qwen 2 [Yang et al., 2024], and Yi 1.5 [Young et al., 2024]. We explore the following applications: (i) benchmark performance prediction for larger models from a specific LLM family, (ii) interpretability of the scaling of skills (can help practitioners allocate resources based on the skills of interest), and (iii) downstream tasks performance prediction.

### 4.1 Data

We expand the dataset made available by Ruan et al. [2024], including more models from the HuggingFace Open LLM leaderboard v1 [Beeching et al., 2023] and v2 [Fourrier et al., 2024]. In our extended dataset, we have a total of 30 families[2], which 28 are on v1 of the Open LLM Leaderboard and 17 families measured on v2 of the Open LLM Leaderboard. Furthermore, there are 15 families at the intersection of the two versions. Furthermore, we collect data and present results on the performance of a variety of instruction-tuned versions of the base models we consider. As far as we are aware, our dataset is the most comprehensive among prior works on benchmark data scaling laws. Please check Appendix G for details on the included models.

### 4.2 Comparing scaling laws in terms of prediction errors

In this section, we compare the predictive power of different scaling laws in predicting LLM performance in all the considered benchmarks; we focus on the two versions of the Open LLM Leaderboard, which include 12 benchmarks: GSM8k [Cobbe et al., 2021], MATH Lvl 5 [Hendrycks et al., 2021], MMLU [Hendrycks et al., 2020], MMLU-PRO [Wang et al., 2024], BBH [Suzgun et al., 2022], GPQA [Rein et al., 2023], MUSR [Sprague et al., 2023], TruthfulQA [Lin et al., 2021], HellaSwag [Zellers et al., 2019], Winogrande [Sakaguchi et al., 2019], ARC [Clark et al., 2018], and IFEval [Zhou et al., 2023]. We apply a leave-one-out cross-validation algorithm to obtain test errors for each family of models. We consider base models and instruct models to belong to distinct

---

[2]If we consider that instruct and base models are from different families, we end up with 53 families.

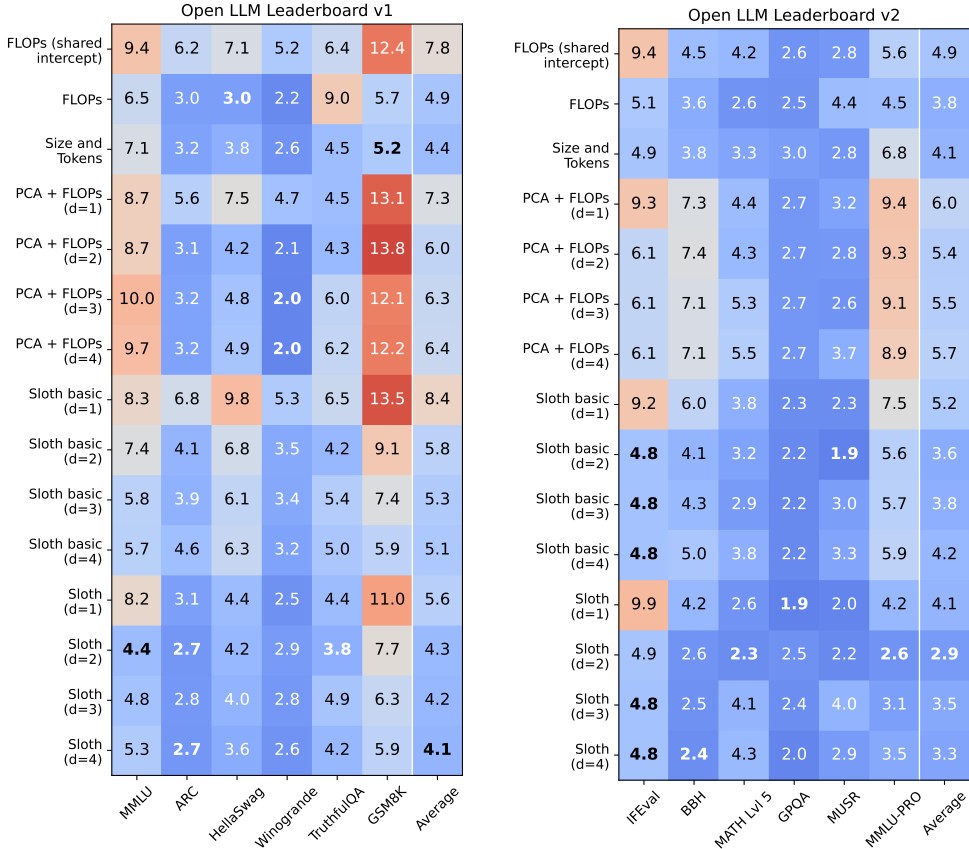

Figure 1: The figure shows the average (across LLM families) mean-absolute-error (MAE) (within a family) for different methods. `Sloth` performs competitively, with errors similar to or lower than the "Size and Tokens" variant, indicating its effective inductive bias.

families (they will not share the same intercept in our model, for example), but we do not include the instruct (resp. base) family in the training set when the corresponding base (resp. instruct) family is in the test set. Moreover, we do not test older versions of recent families if they are available in the training set, *e.g.*, we do not include LLaMa 2 in the set of test families if LLaMa 3 is present in the training set. In this section, we present results for the two leaderboards separately; in Figures 12 and 17 of the Appendix, we also present results for the intersection of the two leaderboards.

In the first set of experiments, we consider the case in which only the smallest model of the test family is observed at training time. Because of that reason, we cannot fit the general scaling law in (2.2) in which both the intercept and slope are family dependent. In this scenario, we consider our main baselines to be (i) the model in (2.2) with shared intercept and slope [Owen, 2024] ("FLOPs (shared intercept)"), (ii) the same model but only with shared slope ("FLOPs"), (iii) a version of the PCA idea[3] proposed by Ruan et al. [2024] in which we predict the principal components using the FLOPs model with shared slope that are then mapped to the benchmark scores ("PCA + FLOPs"), (iv) and our model with trainable activation function but assuming $\Lambda$ is identity ("Size and Tokens"; implies $d = J$). Moreover, we include two versions of `Sloth`. In the "basic" one, we fix $\sigma$ to be sigmoid, and $\gamma_j$'s are given by the $100\%$ over the number of alternatives in the case of multiple-choice benchmarks[4] and 0 otherwise, except for TruthfulQA, which we compute the first percentile of the scores coming from the full Open LLM Leaderboard and fix the lower asymptote to that value. In previous sections, we mentioned that `Sloth` is parameter efficient; we include a parameter count analysis in Appendix F where we compare our model with other well-performing baselines.

---

[3]We include more details about this approach in Appendix E.

[4]When the benchmark has subsections with a different number of alternatives, we compute the asymptote parameters per subsection and then compute an overall asymptote using a weighted average in which the weights are proportional to the number of examples in each subsection.

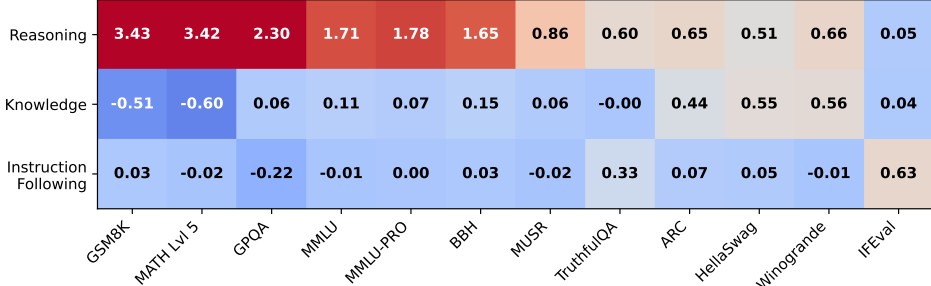

Figure 2: Needed skills for each benchmark. In this figure, we report the estimated loadings Λ and, based on their values, we give them appropriate names.

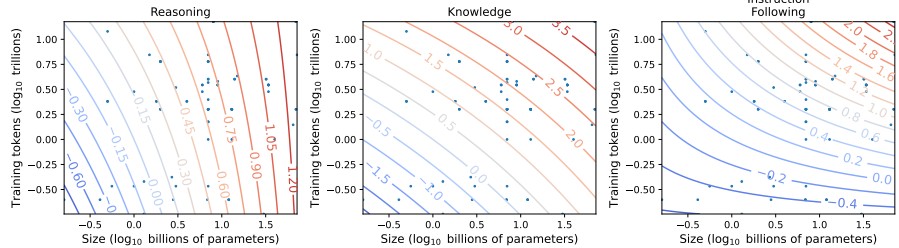

Figure 4: In this figure, we plot the skill levels (output) subtracted by the family-specific intercept terms against inputs in the x and y-axis. From these plots, we can see how each one of the inputs can differently affect the production of skills. For example, "Reasoning" showed to be more affected by model size than tokens when compared to other skills. Moreover, "Knowledge" is more influenced by inputs (level curves are steeper) in general, while the other skills should be more sensitive to other family-dependent factors.

Figure 1 gives the results for the first set of experiments. It depicts the average mean absolute error of all methods when predicting LLM benchmark performance, which is measured in percentage points. It shows the competitiveness of `Sloth` in terms of prediction quality. One important thing to notice is that `Sloth` errors are similar or lower than the "Size and Tokens" variant, suggesting that the assumed low-dimensional structure between benchmarks results is a good inductive bias. We highlight that the analysis includes recent families like LLaMa 3, Qwen 2, and Yi 1.5. For more details on the tested models and extra related results, including model-specific results, please check Appendix H.2. The extra results are qualitatively similar

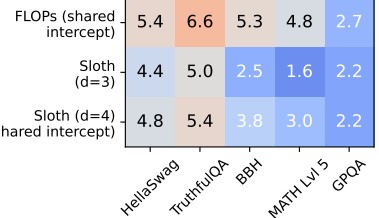

Figure 3: Running `Sloth` with shared intercept can offer a great way to model scaling laws that are family-independent.

to the ones in Figure 1, in which `Sloth` often beats the baselines. In a second set of experiments, we consider the case in which the two smallest models of the test family are observed at training time. In this way, we can fit (2.2) making both parameters family dependent. The results are qualitatively similar to the one presented on Figure 1 and are presented in Appendix H.3. Moreover, we include Figure 10 in the appendix, which is a version of Figure 1 in which Mean Absolute Percentage Error (MAPE) is used instead of MAE; results are qualitatively similar.

In an extra set of experiments, we show that family-specific intercept models are not always needed; we can still get good prediction results for some benchmarks even if we consider a shared intercept between families. The advantage of this approach is that we can claim for a *general* scaling law that holds for all families. Figure 3 shows us a subset[5] of Figure 12 in the appendix and it is built under the same conditions as Figure 1. It is possible to see that, for a subset of benchmarks `Sloth` with shared intercept is a strong alternative to the FLOPs model used by Owen [2024]. In some cases, it gets similar prediction errors relative to more complete versions of `Sloth`.

---

[5]We selected the best $d$ for both versions of `Sloth`.

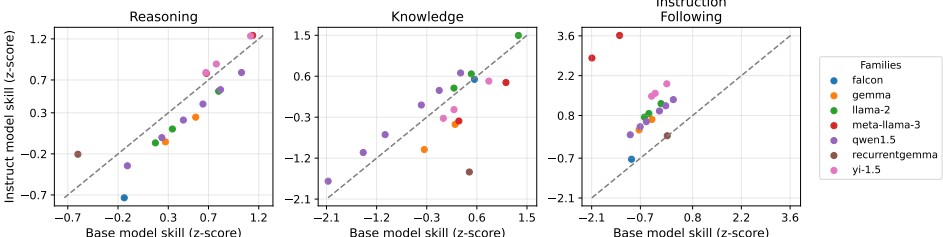

Figure 5: We compare the skills of base (x-axis) and instruction-tuned models (y-axis); if a model lies on the 45-degree line, it means that the model has the same skill level in its base and instruct versions. Gains from instruction tuning (IT) for different families on three latent skills. Findings include a large and positive impact on "Instruction Following" and that provides much larger variations in this skill when compared to inputs seen in Figure 4. Moreover, IT had a moderate and negative effect on "Reasoning" and mixed effects on "Knowledge".

## 4.3 Interpreting the latent skills

In this section, we use the intersection between the two leaderboards, aiming to get more insights from the combined data. Since we have an identifiability result for the "basic" version of `Sloth`, in which we fix the lower asymptotes $\gamma_j$'s and the link function to be sigmoid (see Section A), we opt for interpreting that version of the model. We set $d = 3$ as that model version achieved the best prediction results in Figure 12. Figure 2 illustrates the model loadings, $\Lambda$, from which we assign names to the three dimensions based on our subjective interpretation. We include the loadings for $d = 2$ and $d = 4$ in Appendix I. To complement our exploration, we include Figure 4, which gives us the level curves of different skills (disregarding the family-specific intercept term), and Figure 5 in the appendix that compares the skills of base and instruction-tuned models; in this figure, we include LLM families with more number of models. In both figures, the numbers are given in terms of standard deviations as the skills are standardized to have a zero mean and unitary standard deviation.

**Reasoning skill:** The first dimension, with strong loadings from benchmarks such as GSM8K, MATH, GPQA, MMLU(-PRO), BBH, and MUSR, is labeled "Reasoning." The benchmarks GSM8k and MATH Lvl 5 consist entirely of mathematical word problems while MMLU/MMLU-PRO and GPQA also contain mathematical and advanced science questions. On the other hand, BBH includes logic deduction and linguistic reasoning puzzles. The strong dependence of BBH on the "Reasoning" skill suggests that in language models, there is an association between logical reasoning, general linguistic ability, and mathematical ability. Finally, MuSR is a benchmark that evaluates "multistep soft reasoning tasks specified in a natural language narrative" [Sprague et al., 2023]. Figure 4 shows that Reasoning is primarily a function of model size, with a small dependence on the number of training tokens used. Moreover, the first plot of Figure 5 in the appendix compares base models versus their instruction-tuned versions in terms of Reasoning, and we found that there is no clear rule: instruction tuning can either increase or decrease the ability of an LLM to reason. These findings are robust for different values of $d$ as we can see in the figures of Appendix I.

**Knowledge skill:** The second dimension is positively loaded on ARC, HellaSwag, and Winogrande. These three benchmarks measure the ability of LLMs to remember common sense and basic knowledge; we denominate this skill as "Knowledge". More specifically, ARC consists of grade school-level science questions, HellaSwag is meant for sentence completion for common scenarios, and Winogrande common sense pronoun resolution problems. Contrasting with Reasoning, Figure 4 shows that Knowledge is highly influenced by both model size and number of training tokens. Moreover, we can see that the range of standard deviations in the middle plot is much greater than in the other two plots, giving us evidence that this skill might be more sensitive to increases in compute resources and less dependent on the LLM families themselves. On the other hand, Figure 5 in the appendix does not show any strong evidence of the effect of instruction tuning on this skill. These findings are similar to the ones reported in Appendix I for different values of $d$.

**Instruction following skill:** IFEval, which is positively and heavily loaded in this skill, measures how well language models produce answers that follow a verifiable set of instructions; for example, including a keyword $x$ number of times in responses. Therefore, we call it "Instruction Following". An interesting fact is that instruction tuning has a strong positive effect on this skill for all depicted families we can see in Figure 5 in the appendix. The effect can also be observed in Figure 28 of the appendix. When $d = 2$, instruction following gets mixed with other skills and we are not able to see

this effect. Regarding Figure 4, we see that Instruction Following depends on both model size and tokens. Unfortunately, this interpretation does not hold when $d = 4$ as seen in Appendix I.

## 4.4 Predicting LLM performance on downstream tasks

Another useful application of `Sloth`, which is inspired by Ruan et al. [2024], is to predict the performance in a downstream task for a large model from a relatively small number of prior performance observations from that task. We use `Sloth` to estimate the latent skills of hypothetical LLMs and then use them to predict the performance of those LLMs in downstream tasks. With this approach, we expand on the experiments of Ruan et al. [2024], which do not consider performance prediction of hypothetical LLMs.

The basic prediction pipeline is as follows. First, use standard LLM leaderboards to fit a scaling law for skills using `Sloth`. Second, use existing LLM performance on the downstream task to model how performance can be predicted from latent skills. Third, use `Sloth` to predict the skills of a (hypothetical) LLM of interest, *e.g.*, a larger version of an existing LLM. Finally, use the model fitted in the second step to predict the performance of the hypothetical model in the downstream task.

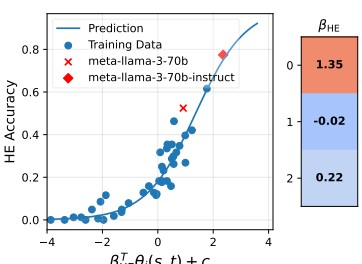

We evaluate this pipeline on two downstream tasks, predicting the performance of `meta-llama-3-70B` and `meta-llama-3-70B-instruct` on code completion and `meta-llama-3-70B-instruct` on emotional intelligence tasks. We fit the same model shown in Section 4.3, but *do not include* `meta-llama-3-70B` *or* `meta-llama-3-70B-instruct` *in the training set* (see Figure 32 for the loadings of the latent skills, which is similar to Figure 2). Next, using either HumanEval [Chen et al., 2021] or EQ bench data [Paech, 2024], we fit a regression model with logistic link using latent skills as features and performance on the downstream task as target. Together, this provides us with sufficient information to predict the performance of the held-out models on both tasks with decent accuracy. Figure 6 depicts this logistic curve and the actual values for HumanEval;

Figure 6: Predicting model performance in complex downstream tasks like code completion ("HE") for LLaMa 3 70B (base/instruct). In the first step, we fit `Sloth` without including LLaMa 3 70B (base/instruct) in the training set. In the second step, we fit a regression model connecting skills and downstream performance. Finally, we predict LLaMa 3 70B (base/instruct) performance from their predicted `Sloth` skills.

EQ bench results can be found in Appendix J. Moreover, we can see that "Reasoning" is by far the most important skill in predicting coding ability while a mixture of "Reasoning" and "Knowledge" is needed for emotional intelligence (see Figure 32 for a more accurate interpretation of the loadings).

In Appendix J, a similar test is provided for agentic capability measured by AgentBench [Liu et al., 2023], although to avoid overfitting, in this case, we must fit `Sloth` with no family-specific intercept.

## 4.5 Predicting performance behavior with scaled inference compute

In this section, we demonstrate how our method, `Sloth`, combined with concepts like Item Response Theory (IRT) [Reckase, 2009], can predict how the performance of an LLM scales with increased inference through repeated sampling [Brown et al., 2024]. For this experiment, we utilize the MATH dataset [Hendrycks et al., 2021], made available by Brown et al. [2024], which evaluates 10 LLMs also included in our dataset. These models are part of the LLaMa 3 Instruct, Gemma, and Pythia families. The process is:

1. Train `Sloth` on our full dataset of 12 benchmarks, excluding the largest models in the LLaMa 3 Instruct, Gemma, and Pythia families.
2. Fit a logistic regression model for each MATH question in Brown et al. [2024]'s data, using the skills $\theta_i(s, t)$ of the 7 training LLMs as covariates to predict the probability of correctly solving those problems.
3. Estimate the skills for the three test models using our scaling law, then predict their probabilities of answering each MATH question correctly via the logistic regressions, and then predict the pass@$k$ metrics for the 3 test models using those probabilities. The predicted pass@$k$ metric for a certain LLM is given by the average of $1 - (1 - \hat{p}_j)^k$'s (across $j$'s) if $\hat{p}_j$ is the predicted probability of the LLM of interest getting question $j$ correct.

Figure 7 illustrates that `Sloth` can accurately predict test-time scaling behavior for these models. Note that unlike scaling law in Brown et al. [2024], `Sloth` can predict inference compute gains for *hypothetical* LLMs before committing resources to training them. This highlights a practical application where practitioners can estimate the potential performance of a hypothetical model at test time, given a computational budget.

### 4.6 Compute-optimal scaling of skills

One relevant question is: given a certain budget in FLOPs, how do we invest it to maximize one skill of interest? This type of analysis is novel for skills and it has only been carried out for validation loss (*e.g.*, by Kaplan et al. [2020], Hoffmann et al. [2022]). A summary of the mathematical formulation of this question is given in the following and exposed in detail in Appendix B. For each model family $i$ and skill $k$, we have

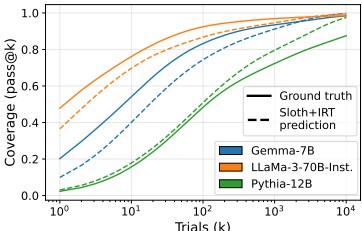

$$\theta_{ik}(s,t) = \alpha_{ik} + \beta_{k0}\log(s) + \beta_{k1}\log(t) + \beta_{k2}\log(s)\log(t).$$

Define $u = \log(s)$, $v = \log(t)$, and $l = \log(c) - \log(6)$. For simplicity, here we consider a simple, but widely used [Kaplan et al., 2020, Hoffmann et al., 2022], compute budget constraint $6st = c$, which is equivalent to $u + v = l$. We add extra constraints on $u$ and $v$ based on the training data support to avoid unreasonable out-of-support predictions. We optimize

Figure 7: `Sloth` can accurately predict test-time scaling behavior for LLaMa 3 Instruct, Gemma, and Pythia models for MATH performance when coupled with Item Response Theory.

$$\max_{u,v} \alpha_{ik} + \beta_{k0}u + \beta_{k1}v + \beta_{k2}uv \quad \text{subject to} \quad u + v = l, \quad u \in [\underline{u}, \overline{u}], \ v \in [\underline{v}, \overline{v}].$$

Substituting $v = l - u$ reduces this to a quadratic function $g_{ik}(u)$, whose maximizer, within the observed range $\mathcal{U} = [\max(l - \overline{v}, \underline{u}), \min(l - \underline{v}, \overline{u})]$, determines the optimal allocation. Our analysis indicates that the optimal values for $s$ and $t$ do not depend on the model family. Table 1 reports our results for instruction following; a table with results for the other two skills is shown in Appendix B.

## 5 Conclusion

In conclusion, we have introduced the `Sloth` scaling laws as a novel approach to predicting the performance of large language models across benchmarks and model families. By leveraging the correlations between benchmark scores and assuming that LLM performance is governed by a set of interpretable, low-dimensional latent skills, our approach offers a more efficient and flexible framework for understanding and predicting LLM behavior. The ability to estimate model performance across a variety of bench-

| FLOPs (1e19) | Instruction Following | |
| --- | --- | --- |
| | Params (B) | Tokens (T) |
| 100 | 0.16 | 1.04 |
| 578 | 0.30 | 3.24 |
| 3346 | 0.72 | 7.78 |
| 19360 | 2.15 | 15.0 |
| 112005 | 12.44 | 15.0 |
| 648000 | 72.0 | 15.0 |

Table 1: Optimal parameter and token allocation.

marks and tasks, even with minimal data from individual model families, highlights the practical utility of `Sloth` scaling laws. Our empirical results demonstrate that `Sloth` can accurately predict the performance of LLMs across multiple benchmarks while providing insights into the relationship between computational resources and model capabilities.

**Limitations.** From the predictive side of `Sloth`, we believe that the main limitation is that the model is still dependent, most of the time, on seeing data from at least one LLM from the LLM family of interest. Moreover, we train the link function in the best version of `Sloth` using flexible neural networks, which can interpolate data very well, but have no guarantee of extrapolation when the (hypothetical) LLM of interest is very different from others in the training set. From the interpretability side, we only understand the identification problems, such as transformations in the latent space, that can arise in a simple instance of `Sloth`: fixed activation function $\sigma$ and $\gamma_j$'s. This fact limits our understanding and interpretability of the most advanced versions of the model.

## 6 Acknowledgements

This paper is supported by the National Science Foundation (NSF) grants DMS-2027737, DMS-2113373, DMS-2414918, and a gift from OpenAI.

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

# A   Identifiability of model parameters and interpretability

To interpret `Sloth` parameters, we need to guarantee they are identifiable. Given that our scaling law models the function $\mu_{ij}(s,t) = \mathbf{E}[Y_{ij}(s,t)]$, that condition is equivalent to the following statement: if two sets of parameters are responsible for characterizing $\mu_{ij}(s,t)$, then those set of parameters should be the same up to predictable variations such as translations or rotations. To prove identifiability, we work with a fixed and invertible $\sigma$, as usually done in the literature, and assume $\gamma_j$'s are fixed. The last condition is reasonable since these constants are usually known beforehand, *e.g.*, it is well accepted that the lower asymptote $\gamma_j$ for MMLU [Hendrycks et al., 2020] performance is $25\%$ which is given by $100\%$ divided by the number of multiple-choice alternatives. Denote our fixed design matrix as $X \in \mathbf{R}^{n \times p}$, where each row is given by an LLM and $p$ equals 3 plus the number of families, and define

$$B = \begin{pmatrix} \beta_1 & \cdots & \beta_d \\ \alpha_{11} & \cdots & \alpha_{1d} \\ \vdots & \ddots & \vdots \\ \alpha_{m1} & \cdots & \alpha_{md} \end{pmatrix} \in \mathbf{R}^{p \times d}$$

such that the rows of $XB \in \mathbf{R}^{n \times d}$ give the skills vectors $\theta^{(i)} \triangleq (XB)^{(i)}$'s of all models in our dataset. Here $n$ denotes the total number of models in the dataset and $m$ is the total number of model families. To prove identifiability, we adopt standard assumptions from the factor analysis literature [Chen et al., 2019] or regression literature, which assumes that the skills vectors $\theta^{(i)} \in \mathbf{R}^{1 \times d}$'s are standardized, *i.e.*, their average is null while their covariance matrix is fixed, rank($\Lambda$) = $d$, and rank($X$) = $p$.

**Assumption A.1** (Identifiability constraints). *Assume that*

$$\frac{1}{n} \sum_{i=1}^{n} \theta^{(i)} = 0, \quad \frac{1}{n} \sum_{i=1}^{n} \theta^{(i)\top} \theta^{(i)} = \Psi,$$

*and that rank($\Lambda$) = $d$ and rank($X$) = $p$, where $\theta^{(i)}$ denotes the $i$-th row of $XB$ and $\Psi$ is a positive definite matrix.*

One possible choice for the covariance matrix is $\Psi = I_d$ [Chen et al., 2019], which assumes uncorrelated skills. One implicit implication of Assumption A.1 is that $n \geq p \geq d$ must be satisfied, otherwise the covariance matrix cannot be full rank. This condition is satisfied in our experiments. Under Assumption A.1, we show the identifiability of the model parameters up to a transformation of $\Lambda$ tied to a transformation of $B$, which leaves the outputs of the model unchanged. This means that we can potentially approximate the true values for $\Lambda$ and $B$ up to a transformation, which is usually the norm within the class of exploratory factor analysis models.

**Theorem A.2.** *Given that the true set of model parameters is $(\Lambda, b, B)$, if there is another set of parameters $(\tilde{\Lambda}, \tilde{b}, \tilde{B})$ that satisfy*

$$\sigma\left(\Lambda(XB)^{(i)\top} + b\right) = \sigma\left(\tilde{\Lambda}(XB)^{(i)\top} + \tilde{b}\right) \text{ for all } i \in [n],$$

*then, under the Assumption A.1, we have $\tilde{b} = b$, $\tilde{\Lambda} = \Lambda M$, and $\tilde{B} = B(M^\top)^{-1}$ for an invertible matrix $M \in \mathbf{R}^{d \times d}$. In particular, $M$ is orthogonal if $\Psi = I_d$, i.e., $M^\top M = MM^\top = I_d$.*

*Proof.* We start proving that $b = \tilde{b}$. Because $\sigma$ is invertible, we get

$$\Lambda(XB)^{(i)\top} + b = \tilde{\Lambda}(X\tilde{B})^{(i)\top} + \tilde{b} \text{ for all } i \in [n],$$

and consequently by the standardization of the latent skills

$$\Lambda \left[\frac{1}{n} \sum_{i=1}^{n} (XB)^{(i)\top}\right] + b = \tilde{\Lambda} \left[\frac{1}{n} \sum_{i=1}^{n} (X\tilde{B})^{(i)\top}\right] + \tilde{b} \Rightarrow b = \tilde{b}.$$

Now, we prove that $\tilde{\Lambda} = \Lambda M$. Given that $b = \tilde{b}$, we have

$$\Lambda(XB)^{(i)\top} = \tilde{\Lambda}(X\tilde{B})^{(i)\top} \text{ for all } i \in [n],$$

and consequently by the standardization of the latent skills

$$\Lambda \left[ \frac{1}{n} \sum_{i=1}^{n} (XB)^{(i)\top} (XB)^{(i)} \right] \Lambda^\top = \tilde{\Lambda} \left[ \frac{1}{n} \sum_{i=1}^{n} (X\tilde{B})^{(i)\top} (X\tilde{B})^{(i)} \right] \tilde{\Lambda}^\top \Rightarrow \Lambda \Psi \Lambda^\top = \tilde{\Lambda} \Psi \tilde{\Lambda}^\top.$$

By Cholesky's decomposition, we can write $\Psi = LL^\top$, for a lower triangular matrix $L$. If we define $\Lambda' \triangleq \Lambda L$ and $\tilde{\Lambda}' \triangleq \tilde{\Lambda} L$, then

$$\Lambda' \Lambda'^\top = \tilde{\Lambda}' \tilde{\Lambda}'^\top.$$

Because $\text{rank}(\Lambda) = d$, we have that $\text{rank}(\Lambda') = d$ and we claim that $\tilde{\Lambda}' = \Lambda' U$ for an orthogonal matrix $U \in \mathbf{R}^{d \times d}$. To see that, first, realize that

- $\text{rank}(\Lambda') = \text{rank}(\Lambda'\Lambda'^\top) = \text{rank}(\tilde{\Lambda}'\tilde{\Lambda}'^\top) = \text{rank}(\tilde{\Lambda}')$. We see this by realizing that the null spaces of $\Lambda'^\top$ and $\Lambda'\Lambda'^\top$ are the same: for an arbitrary vector $z$, $\Lambda'^\top z = 0 \Rightarrow \Lambda'\Lambda'^\top z = 0$ and $\Lambda'\Lambda'^\top z = 0 \Rightarrow \Lambda'^\top \Lambda'\Lambda'^\top z = 0 \Rightarrow \Lambda'^\top z = 0$, where the last implication follows from the assumption that $\Lambda'^\top \Lambda'$ is full rank ($\text{rank}(\Lambda') = d$). Because the null spaces of $\Lambda'^\top$ and $\Lambda'\Lambda'^\top$ are the same, their ranks should be the same as well. The same reasoning applies to $\tilde{\Lambda}'\tilde{\Lambda}'^\top$ and $\tilde{\Lambda}'$, proving this intermediate result.

- Because $\Lambda'$ and $\Lambda'\Lambda'^\top$ have the same rank, the column space of these two matrices must be the same as the columns of $\Lambda'\Lambda'^\top$ are given by linear combinations of columns of $\Lambda'$. Same for $\tilde{\Lambda}'$ and $\tilde{\Lambda}'\tilde{\Lambda}'^\top$. Consequently, the column spaces of $\Lambda'$ and $\tilde{\Lambda}'$ are the same.

Because the column spaces of $\Lambda'$ and $\tilde{\Lambda}'$ are the same, there must be an invertible matrix $U$ such that $\tilde{\Lambda}' = \Lambda' U$. But then

$$\Lambda' \Lambda'^\top = \tilde{\Lambda}' \tilde{\Lambda}'^\top = \Lambda' U U^\top \Lambda'^\top \Rightarrow \Lambda'^\top \Lambda' \Lambda'^\top \Lambda' = \Lambda'^\top \Lambda' U U^\top \Lambda'^\top \Lambda' \Rightarrow$$

$$\Rightarrow U U^\top = (\Lambda'^\top \Lambda')^{-1} (\Lambda'^\top \Lambda')(\Lambda'^\top \Lambda')(\Lambda'^\top \Lambda')^{-1} = I$$

and

$$U U^\top = I \Rightarrow U^\top U U^\top U = U^\top U \Rightarrow U^\top U = I$$

Because $\tilde{\Lambda}' = \Lambda' U$, we have that

$$\tilde{\Lambda} L = \Lambda L U \Rightarrow \tilde{\Lambda} = \Lambda L U L^{-1} = \Lambda M.$$

If $\Psi = I_d$, then $L = I_d$ and $M = U$.

Finally, we prove that $\tilde{B} = B(M^\top)^{-1}$. From previous considerations, we can write

$$\Lambda B^\top X^\top = \Lambda M \tilde{B}^\top X^\top \Rightarrow \Lambda^\top \Lambda B^\top X^\top = \Lambda^\top \Lambda M \tilde{B}^\top X^\top$$

$$\stackrel{\text{rank}(\Lambda)=d}{\Rightarrow} XB = X\tilde{B}M^\top$$

$$\Rightarrow X^\top X B = X^\top X \tilde{B} M^\top$$

$$\stackrel{\text{rank}(X)=p}{\Rightarrow} B = \tilde{B}M^\top$$

$$\Rightarrow \tilde{B} = B(M^\top)^{-1}$$

If $\Psi = I_d$, then $L = I_d$ and $(M^\top)^{-1} = U$. $\qquad\qquad\qquad\qquad\qquad\qquad\qquad\qquad\qquad\square$

From our proof, we can see that the matrix $M$ is dependent on the specification of $\Psi$.

## A.1 Interpretability

In practical situations, it is hard to fix the covariance matrix of skills to something meaningful before fitting the model, as suggested in Section A.1. To make the model interpretable, we mirror a standard approach used in factor analysis, *e.g.*, in Chen et al. [2019]'s applications. First, we fit Sloth without any constraints on the covariance of skills obtaining the estimates $(\hat{\Lambda}, \hat{b}, \hat{B})$. Second, we find the matrix $A \in \mathbf{R}^{d \times d}$ such that the skills $X\hat{B}A$ have covariance identity, update $\hat{B} \leftarrow \hat{B}A$, and update $\hat{\Lambda} \leftarrow \hat{\Lambda}(A^\top)^{-1}$ so the model outputs remains unchanged, because $\hat{\Lambda}(X\hat{B})^\top = \hat{\Lambda}(A^\top)^{-1}(X\hat{B}A)^\top$.

Third, we find a matrix $M \in \mathbf{R}^{d \times d}$ such that $\hat{\Lambda}M$ is easily interpretable (*e.g.*, it is a sparse matrix); there are different methods to find $M$ and, in this paper, we use the *Geomin* [Yates, 1987, Chen et al., 2019] oblique rotation method to find a suitable $M$ using the Python package FactorAnalyzer [Biggs, 2019]. We then update $\hat{\Lambda} \leftarrow \hat{\Lambda}M$ and, to make the model invariant, we also update $\hat{B} \leftarrow \hat{B}(M^\top)^{-1}$; the resulting skills are still guaranteed to have unitary standard deviations, so their covariance equals their correlation. Finally, we standardize the columns of the skills $X\hat{B}$ to have zero mean, while keeping the correlation structure unchanged. This last step implies that $\hat{b}$ must be translated to make the model invariant.

## B    Compute-optimal scaling

In this section, we derive compute-optimal scaling rules for skills. Specifically, given a language model family and a particular skill, our goal is to determine the model configuration that maximizes performance under a fixed computation budget $6st = c$.

Consider a model family $i$ with skill $k$ defined as

$$\theta_{ik}(s, t) = \alpha_{ik} + \beta_{k0} \log(s) + \beta_{k1} \log(t) + \beta_{k2} \log(s) \log(t).$$

Letting $u = \log(s)$, $v = \log(t)$, and $l = \log(c) - \log(6)$, we formulate the optimization problem as:

$$\max_{u,v} \alpha_{ik} + \beta_{k0}u + \beta_{k1}v + \beta_{k2}uv \quad \text{s.t.} \quad u + v = l.$$

This reduces to a simpler optimization problem in terms of $u$ alone:

$$\max_u g_{ik}(u), \quad \text{where} \quad g_{ik}(u) \triangleq -\beta_{k2}u^2 + (\beta_{k0} - \beta_{k1} + \beta_{k2}l)u + (\alpha_{ik} + \beta_{k1}l).$$

To prevent our compute-optimal scaling method from making unreasonable predictions, we further restrict its solution to the ranges of $u$ and $v$ observed in the training data. Hence, we impose constraints $u \in [\underline{u}, \overline{u}]$ and $v = l - u \in [\underline{v}, \overline{v}]$, where bounds $\underline{u}, \overline{u}, \underline{v}, \overline{v}$ are set based on quantiles from training data. Combining these constraints yields:

$$u \in \mathcal{U} \triangleq [\max(l - \overline{v}, \underline{u}), \min(l - \underline{v}, \overline{u})].$$

Thus, the optimization problem becomes:

$$\max_{u \in \mathcal{U}} g_{ik}(u),$$

which is straightforward to solve. Specifically, if $\beta_{k2} > 0$, $g_{ik}(u)$ is a concave parabola, and the maximizer is either at the vertex (if it falls within $\mathcal{U}$) or at one of the interval endpoints $\max(l - \overline{v}, \underline{u})$ or $\min(l - \underline{v}, \overline{u})$.

Table 2 extends the results in Table 1 for all skills.

| FLOPs (1e19) | Reasoning | | Knowledge | | Instruction Following | |
|---|---|---|---|---|---|---|
| | Params (B) | Tokens (T) | Params (B) | Tokens (T) | Params (B) | Tokens (T) |
| 100 | 0.93 | 0.18 | 0.16 | 1.04 | 0.16 | 1.04 |
| 578 | 5.36 | 0.18 | 0.16 | 6.02 | 0.30 | 3.24 |
| 3346 | 30.98 | 0.18 | 0.37 | 15.0 | 0.72 | 7.78 |
| 19360 | 72.0 | 0.45 | 2.15 | 15.0 | 2.15 | 15.0 |
| 112005 | 72.0 | 2.59 | 12.44 | 15.0 | 12.44 | 15.0 |
| 648000 | 72.0 | 15.0 | 72.0 | 15.0 | 72.0 | 15.0 |

Table 2: Optimal allocation of parameters (B) and tokens (T) across skills for various compute budgets (FLOPs).

## C    Connections with factor analysis

Sloth is heavily inspired by (exploratory) factor analysis models. Factor analysis is a statistical technique used to identify underlying relationships between observed variables by reducing the data's

dimensionality [Bishop and Nasrabadi, 2006, Chen et al., 2019]. It assumes that multiple observed variables are influenced by a smaller number of unobserved/latent variables called factors (skills $\theta^{(i)}$, in our case). These factors help explain the correlations among the observed variables. The method aims to model the observed variability and reveal the structure behind the data by estimating the factor loadings ($\Lambda$, in our case). The classical model assumes

$$Y_i = \Lambda\theta_i + b + \varepsilon_i,$$

where $Y_i$ is a vector of variables of interest and $\varepsilon_i$ is an error term. There are versions for the factor model in which a nonlinear model for $Y_i$ is assumed, *e.g.*, in item response theory (IRT) [Reckase, 2006, Chen et al., 2019]. It is usually the case that $\theta_i$ is estimated using a random effects model, *i.e.*, practitioners place a prior distribution on $\theta_i$. In our work, we assume $\theta_i$ is given by a function of observable covariates and a family-specific intercept, which is fitted using data.

## D    Motivating the interaction term in `Sloth`

As shown in Section 3, we include an interaction term between $\log(s)$ and $\log(t)$. In the first place, we consider this as a natural extension of the model that depends on $s$ and $t$ only through FLOPs, since we recover that formulation if $\beta_{k1} = \beta_{k2}$ and $\beta_{k3} = 0$. In the second place, we believe that the dependence of benchmark performances on $\log(s)$ depends on $\log(t)$ (and possibly vice-versa). To motivate this idea we show some plots for two benchmarks we use: MMLU-PRO and BBH. For these plots, we only keep families with a higher number of models. First, realize that in both Figures 8 and 9, the performance within families in the middle plot can be well approximated by a line. Also, the slope of this line has a strong relation with the number of tokens in the last plots. For example, Pythia was trained in a small dataset and its (hypothetical) slopes on the second plot are almost zero in both cases. On the other hand, Qwen2 was trained on more data and its (hypothetical) slope on the middle plots is high. Certainly, this relationship does not always exist, but adding an interaction term in our model helps us to leverage this pattern when it exists.

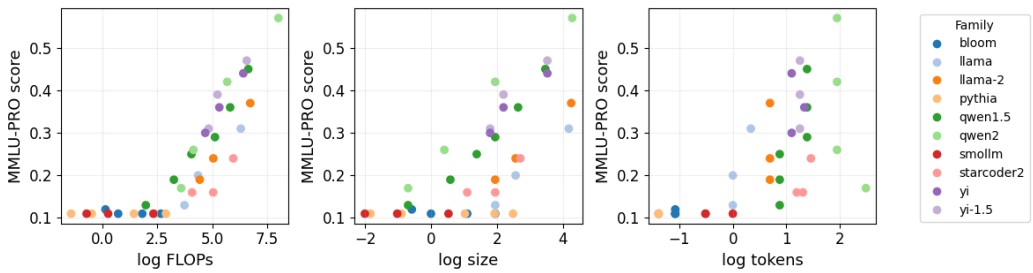

Figure 8: Inputs vs MMLU-PRO scores.

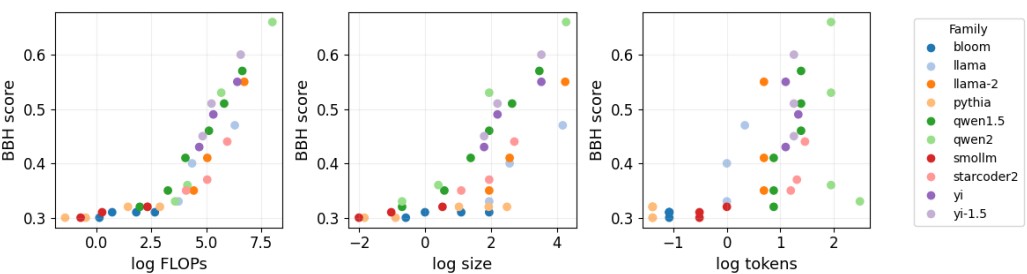

Figure 9: Inputs vs BBH scores.

# E  PCA approach formulation

We follow the ideas of Ruan et al. [2024] as closely as possible to create a prediction method. Moreover, we follow their code[6] and apply PCA with the same set of hyperparameters. Assume we have a matrix of scores $Y \in [0,1]^{n \times J}$ in which columns represent benchmarks and each row represents a language model. We compute the covariance matrix of benchmark scores using $Y$ and then compute its eigenvector matrix $U$, where the columns give the ordered eigenvectors (from the highest eigenvalue to the lowest one). To reduce the dimensions of $Y$, we keep only the first $d$ columns of $YU$, resulting in $\tilde{Y} \in \mathbf{R}^{n \times d}$. For each column of $\tilde{Y}$ (principal components; PCs), we train a linear regression model using logFLOPs as the covariate; in this case, either the intercept or both the intercept and slope can be family-dependent. At test time, we predict the PCs of a held-out model and then go back to the original coordinate axis to obtain the final predictions by computing $\sum_{j=1}^{d} \hat{\mathrm{PC}}_j U_{\cdot,j} \in \mathbf{R}^J$.

# F  Sloth is parameter-efficient: parameter count analysis

When the number of model families $f$ is moderately large, Sloth actually uses **fewer** parameters than the top-performing baselines (both of which consider family-specific intercept or train the activation function). This is because of the assumed latent skill structure of Sloth. For example, with $d = 3$ and we use $\mathcal{J} = 12$ benchmarks:

- Sloth: $12 \cdot (d + 2) + d \cdot (f + 3) = 69 + 3f$ parameters
- "FLOPs" baseline: $12 \cdot (f + 3) = 36 + 12f$ parameters
- "Size and Tokens" baseline: $12 \cdot (f + 5) = 50 + 12f$ parameters

So for $f \geq 4$, Sloth uses fewer parameters than either baseline.

---

[6]See https://github.com/ryoungj/ObsScaling.

# G Models in our dataset

Table 3 gives a detailed view of our dataset. The column "Family" considers that base and instruct models are from different families, while "OriginalFamily" does not. The column "TestFamily" tells if that specific family is considered to be part of the test set in our experiment while the remaining three columns tell if the data is available for these specific benchmarks. For the EQ data, only the following models are available 'gemma-7b-it', 'llama-2-13b-chat', 'llama-2-70b-chat', 'llama-2-7b-chat', 'meta-llama-3-70b-instruct', 'meta-llama-3-8b-instruct', 'qwen1.5-1.8b-chat', 'qwen1.5-14b-chat', 'qwen1.5-32b-chat', 'qwen1.5-4b-chat', 'qwen1.5-7b-chat', 'yi-1.5-34b-chat', 'yi-1.5-6b-chat', 'yi-1.5-9b-chat', 'yi-34b-chat'.

|    | Model | Family | OriginalFamily | TestFamily | Leaderboard1 | Leaderboard2 | HumanEval |
|----|-------|--------|----------------|------------|--------------|--------------|-----------|
| 0  | bloom | bloom | bloom | True | True | False | True |
| 1  | bloom-1b1 | bloom | bloom | True | True | True | True |
| 2  | bloom-3b | bloom | bloom | True | True | True | True |
| 3  | bloom-560m | bloom | bloom | True | True | True | True |
| 4  | bloom-7b1 | bloom | bloom | True | True | True | True |
| 5  | blossom-v5.1-34b | blossom-v5.1 | yi-1.5 | False | True | True | False |
| 6  | blossom-v5.1-9b | blossom-v5.1 | yi-1.5 | False | False | True | False |
| 7  | codegen-16b-nl | codegen-nl | codegen | True | True | False | True |
| 8  | codegen-6b-nl | codegen-nl | codegen | True | True | False | True |
| 9  | codellama-13b | codellama | codellama | True | True | False | True |
| 10 | codellama-34b | codellama | codellama | True | True | False | True |
| 11 | codellama-70b | codellama | codellama | True | True | False | True |
| 12 | codellama-7b | codellama | codellama | True | True | False | True |
| 13 | deepseek-coder-1.3b-base | deepseek-coder-base | deepseek-coder | True | True | False | True |
| 14 | deepseek-coder-33b-base | deepseek-coder-base | deepseek-coder | True | True | False | True |
| 15 | deepseek-coder-6.7b-base | deepseek-coder-base | deepseek-coder | True | True | False | True |
| 16 | dolly-v2-12b | dolly-v2 | pythia | True | True | True | True |
| 17 | dolly-v2-3b | dolly-v2 | pythia | True | False | True | False |
| 18 | dolly-v2-7b | dolly-v2 | pythia | True | True | True | False |
| 19 | dolphin-2.9.1-yi-1.5-34b | dolphin-2.9.1-yi-1.5 | yi-1.5 | True | True | True | False |
| 20 | dolphin-2.9.1-yi-1.5-9b | dolphin-2.9.1-yi-1.5 | yi-1.5 | True | True | True | False |
| 21 | dolphin-2.9.2-qwen2-72b | dolphin-2.9.2-qwen2 | qwen2 | True | False | True | False |
| 22 | dolphin-2.9.2-qwen2-7b | dolphin-2.9.2-qwen2 | qwen2 | True | False | True | False |
| 23 | falcon-180b | falcon | falcon | True | True | False | False |
| 24 | falcon-40b | falcon | falcon | True | True | True | False |
| 25 | falcon-40b-instruct | falcon-instruct | falcon | True | False | True | False |
| 26 | falcon-7b | falcon | falcon | True | True | True | False |
| 27 | falcon-7b-instruct | falcon-instruct | falcon | True | True | True | False |
| 28 | gemma-2-2b | gemma-2 | gemma-2 | True | False | True | False |
| 29 | gemma-2-2b-it | gemma-2-it | gemma-2 | True | False | True | False |
| 30 | gemma-2-9b | gemma-2 | gemma-2 | True | False | True | False |
| 31 | gemma-2-9b-it | gemma-2-it | gemma-2 | True | False | True | False |
| 32 | gemma-2b | gemma | gemma | True | True | True | True |
| 33 | gemma-2b-it | gemma-it | gemma | True | True | True | True |
| 34 | gemma-7b | gemma | gemma | True | True | True | True |
| 35 | gemma-7b-it | gemma-it | gemma | True | True | True | True |
| 36 | gpt-j-6b | gpt-j-neo-neox | gpt-neo/j | True | True | False | True |
| 37 | gpt-neo-1.3b | gpt-j-neo-neox | gpt-neo/j | True | True | True | True |
| 38 | gpt-neo-125m | gpt-j-neo-neox | gpt-neo/j | True | True | False | True |
| 39 | gpt-neo-2.7b | gpt-j-neo-neox | gpt-neo/j | True | True | True | True |
| 40 | gpt-neox-20b | gpt-j-neo-neox | gpt-neo/j | True | True | False | True |
| 41 | internlm2-20b | internlm2 | internlm2 | True | True | False | False |
| 42 | internlm2-7b | internlm2 | internlm2 | True | True | False | False |
| 43 | llama-13b | llama | llama | False | True | True | True |
| 44 | llama-2-13b | llama-2 | llama-2 | False | True | True | True |
| 45 | llama-2-13b-chat | llama-2-chat | llama-2 | False | True | True | True |
| 46 | llama-2-70b | llama-2 | llama-2 | False | True | True | True |
| 47 | llama-2-70b-chat | llama-2-chat | llama-2 | False | True | True | True |
| 48 | llama-2-7b | llama-2 | llama-2 | False | True | True | True |
| 49 | llama-2-7b-chat | llama-2-chat | llama-2 | False | True | True | True |
| 50 | llama-3-sauerkrautlm-70b-instruct | llama-3-sauerkrautlm-instruct | meta-llama-3 | True | False | True | False |
| 51 | llama-3-sauerkrautlm-8b-instruct | llama-3-sauerkrautlm-instruct | meta-llama-3 | True | True | True | False |

| 52 | llama-30b | llama | llama | False | True | False | True |
|---|---|---|---|---|---|---|---|
| 53 | llama-65b | llama | llama | False | True | True | True |
| 54 | llama-7b | llama | llama | False | True | True | True |
| 55 | meta-llama-3-70b | meta-llama-3 | meta-llama-3 | True | True | True | True |
| 56 | meta-llama-3-70b-instruct | meta-llama-3-instruct | meta-llama-3 | True | True | True | True |
| 57 | meta-llama-3-8b | meta-llama-3 | meta-llama-3 | True | True | True | True |
| 58 | meta-llama-3-8b-instruct | meta-llama-3-instruct | meta-llama-3 | True | True | True | True |
| 59 | mpt-30b | mpt | mpt | True | True | False | True |
| 60 | mpt-30b-chat | mpt-chat | mpt | True | True | False | False |
| 61 | mpt-30b-instruct | mpt-instruct | mpt | True | True | False | False |
| 62 | mpt-7b | mpt | mpt | True | True | False | True |
| 63 | mpt-7b-chat | mpt-chat | mpt | True | True | False | False |
| 64 | mpt-7b-instruct | mpt-instruct | mpt | True | True | False | False |
| 65 | olmo-1b | olmo | olmo | True | True | True | False |
| 66 | olmo-7b | olmo | olmo | True | True | True | False |
| 67 | open_llama_13b | open_llama_ | openllama | False | True | False | False |
| 68 | open_llama_3b | open_llama_ | openllama | False | True | False | False |
| 69 | open_llama_3b_v2 | open_llama__v2 | openllamav2 | False | True | False | False |
| 70 | open_llama_7b | open_llama_ | openllama | False | True | False | False |
| 71 | open_llama_7b_v2 | open_llama__v2 | openllamav2 | False | True | False | False |
| 72 | openhermes-13b | openhermes | llama-2 | False | True | True | False |
| 73 | openhermes-7b | openhermes | llama-2 | False | True | True | False |
| 74 | opt-1.3b | opt | opt | True | True | True | True |
| 75 | opt-125m | opt | opt | True | True | False | True |
| 76 | opt-13b | opt | opt | True | True | False | True |
| 77 | opt-2.7b | opt | opt | True | True | False | True |
| 78 | opt-30b | opt | opt | True | True | True | True |
| 79 | opt-350m | opt | opt | True | True | False | True |
| 80 | opt-6.7b | opt | opt | True | True | False | True |
| 81 | opt-66b | opt | opt | True | True | False | True |
| 82 | orca-2-13b | orca-2 | llama-2 | False | True | True | False |
| 83 | orca-2-7b | orca-2 | llama-2 | False | True | True | False |
| 84 | orca_mini_v3_13b | orca_mini_v3_ | llama-2 | False | True | True | False |
| 85 | orca_mini_v3_70b | orca_mini_v3_ | llama-2 | False | False | True | False |
| 86 | orca_mini_v3_7b | orca_mini_v3_ | llama-2 | False | True | True | False |
| 87 | orca_mini_v7_72b | orca_mini_v7_ | qwen2 | False | False | True | False |
| 88 | orca_mini_v7_7b | orca_mini_v7_ | qwen2 | False | False | True | False |
| 89 | pythia-1.4b | pythia | pythia | True | True | False | True |
| 90 | pythia-12b | pythia | pythia | True | True | True | True |
| 91 | pythia-160m | pythia | pythia | True | True | True | True |
| 92 | pythia-1b | pythia | pythia | True | True | False | True |
| 93 | pythia-2.8b | pythia | pythia | True | True | True | True |
| 94 | pythia-410m | pythia | pythia | True | True | True | True |
| 95 | pythia-6.9b | pythia | pythia | True | True | True | True |
| 96 | pythia-70m | pythia | pythia | True | True | False | True |
| 97 | qwen-14b | qwen | qwen | False | True | False | True |
| 98 | qwen-72b | qwen | qwen | False | True | False | True |
| 99 | qwen-7b | qwen | qwen | False | True | False | True |
| 100 | qwen1.5-0.5b | qwen1.5 | qwen1.5 | False | True | True | True |
| 101 | qwen1.5-0.5b-chat | qwen1.5-chat | qwen1.5 | False | True | True | False |
| 102 | qwen1.5-1.8b | qwen1.5 | qwen1.5 | False | True | True | True |
| 103 | qwen1.5-1.8b-chat | qwen1.5-chat | qwen1.5 | False | True | True | False |
| 104 | qwen1.5-14b | qwen1.5 | qwen1.5 | False | True | True | True |
| 105 | qwen1.5-14b-chat | qwen1.5-chat | qwen1.5 | False | True | True | False |
| 106 | qwen1.5-32b | qwen1.5 | qwen1.5 | False | True | True | True |
| 107 | qwen1.5-32b-chat | qwen1.5-chat | qwen1.5 | False | True | True | False |
| 108 | qwen1.5-4b | qwen1.5 | qwen1.5 | False | True | True | True |
| 109 | qwen1.5-4b-chat | qwen1.5-chat | qwen1.5 | False | True | True | False |
| 110 | qwen1.5-72b | qwen1.5 | qwen1.5 | False | True | False | True |
| 111 | qwen1.5-72b-chat | qwen1.5-chat | qwen1.5 | False | True | False | True |
| 112 | qwen1.5-7b | qwen1.5 | qwen1.5 | False | True | True | True |
| 113 | qwen1.5-7b-chat | qwen1.5-chat | qwen1.5 | False | True | True | False |
| 114 | qwen2-0.5b | qwen2 | qwen2 | True | True | True | False |
| 115 | qwen2-0.5b-instruct | qwen2-instruct | qwen2 | True | False | True | False |
| 116 | qwen2-1.5b | qwen2 | qwen2 | True | True | True | False |
| 117 | qwen2-1.5b-instruct | qwen2-instruct | qwen2 | True | False | True | False |
| 118 | qwen2-72b | qwen2 | qwen2 | True | True | True | False |
| 119 | qwen2-72b-instruct | qwen2-instruct | qwen2 | True | False | True | False |
| 120 | qwen2-7b | qwen2 | qwen2 | True | True | True | False |
| 121 | qwen2-7b-instruct | qwen2-instruct | qwen2 | True | False | True | False |
| 122 | rwkv-4-14b-pile | rwkv-4-pile | rwkv | True | True | False | False |
| 123 | rwkv-4-169m-pile | rwkv-4-pile | rwkv | True | True | False | False |
| 124 | rwkv-4-1b5-pile | rwkv-4-pile | rwkv | True | True | False | False |
| 125 | rwkv-4-3b-pile | rwkv-4-pile | rwkv | True | True | False | False |
| 126 | rwkv-4-430m-pile | rwkv-4-pile | rwkv | True | True | False | False |
| 127 | rwkv-4-7b-pile | rwkv-4-pile | rwkv | True | True | False | False |
| 128 | sauerkrautlm-gemma-2b | sauerkrautlm-gemma | gemma | True | True | True | False |

| | | | | | | | |
|---|---|---|---|---|---|---|---|
| 129 | sauerkrautlm-gemma-7b | sauerkrautlm-gemma | gemma | True | True | True | False |
| 130 | smollm-1.7b | smollm | smollm | True | False | True | False |
| 131 | smollm-1.7b-instruct | smollm-instruct | smollm | True | False | True | False |
| 132 | smollm-135m | smollm | smollm | True | False | True | False |
| 133 | smollm-135m-instruct | smollm-instruct | smollm | True | False | True | False |
| 134 | smollm-360m | smollm | smollm | True | False | True | False |
| 135 | smollm-360m-instruct | smollm-instruct | smollm | True | False | True | False |
| 136 | stablelm-base-alpha-3b | stablelm-base-alpha | stablelm | True | True | False | False |
| 137 | stablelm-base-alpha-7b | stablelm-base-alpha | stablelm | True | True | False | False |
| 138 | starcoder2-15b | starcoder2 | starcoder2 | True | True | True | True |
| 139 | starcoder2-3b | starcoder2 | starcoder2 | True | True | True | True |
| 140 | starcoder2-7b | starcoder2 | starcoder2 | True | True | True | True |
| 141 | starcoderbase | starcoderbase | starcoder | False | True | False | True |
| 142 | starcoderbase-1b | starcoderbase | starcoder | False | True | False | True |
| 143 | starcoderbase-3b | starcoderbase | starcoder | False | True | False | True |
| 144 | starcoderbase-7b | starcoderbase | starcoder | False | True | False | True |
| 145 | wizardlm-13b-v1.0 | wizardlm-v1.0 | llama-2 | False | False | True | False |
| 146 | wizardlm-70b-v1.0 | wizardlm-v1.0 | llama-2 | False | False | True | False |
| 147 | xglm-1.7b | xglm | xglm | True | True | False | True |
| 148 | xglm-4.5b | xglm | xglm | True | True | False | True |
| 149 | xglm-564m | xglm | xglm | True | True | False | True |
| 150 | xglm-7.5b | xglm | xglm | True | True | False | True |
| 151 | yi-1.5-34b | yi-1.5 | yi-1.5 | True | True | True | False |
| 152 | yi-1.5-34b-chat | yi-1.5-chat | yi-1.5 | True | True | True | False |
| 153 | yi-1.5-6b | yi-1.5 | yi-1.5 | True | True | True | False |
| 154 | yi-1.5-6b-chat | yi-1.5-chat | yi-1.5 | True | True | True | False |
| 155 | yi-1.5-9b | yi-1.5 | yi-1.5 | True | True | True | False |
| 156 | yi-1.5-9b-chat | yi-1.5-chat | yi-1.5 | True | True | True | False |
| 157 | yi-34b | yi | yi | False | True | True | True |
| 158 | yi-34b-200k | yi-200k | yi-200k | False | True | False | False |
| 159 | yi-34b-chat | yi-chat | yi | False | True | False | False |
| 160 | yi-6b | yi | yi | False | True | True | True |
| 161 | yi-6b-200k | yi-200k | yi-200k | False | True | False | False |
| 162 | yi-6b-chat | yi-chat | yi | False | False | True | False |
| 163 | yi-9b | yi | yi | False | True | True | False |

# H    Extra performance prediction results

In this section, we present the full versions of the figures presented in the main text and some other extra results.

## H.1    Mean Absolute Percentage Error (MAPE) plot

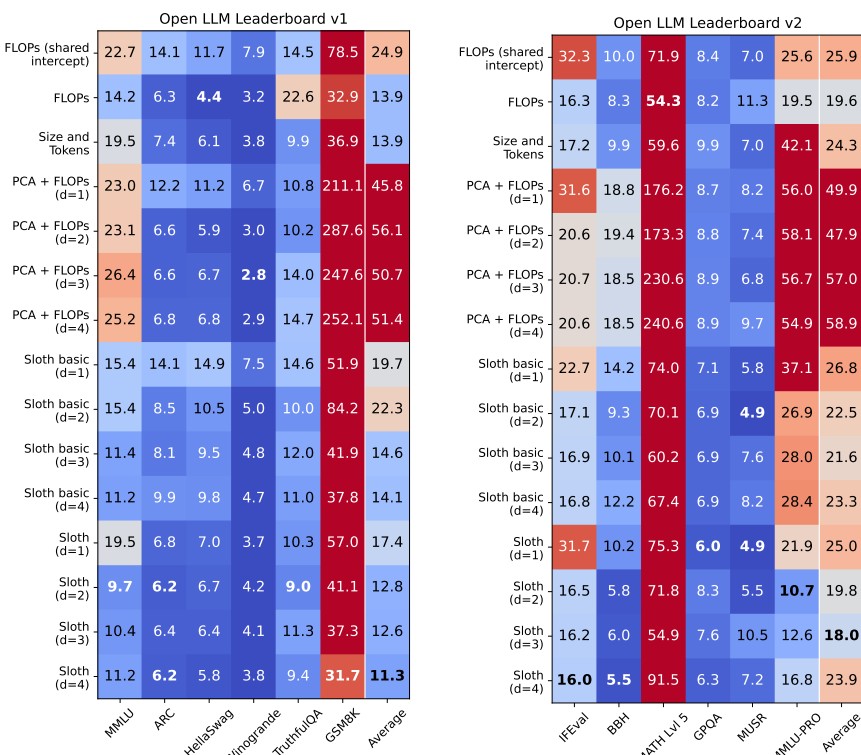

Figure 10: MAPE version of Figure 1. The results are given in percentage points, and we can see that the results are qualitatively similar to the MAE version, with `Sloth` producing the best predictions for both Open LLM Leaderboards.

## H.2 Test families have exactly one model in the training set

### H.2.1 Average prediction loss across models

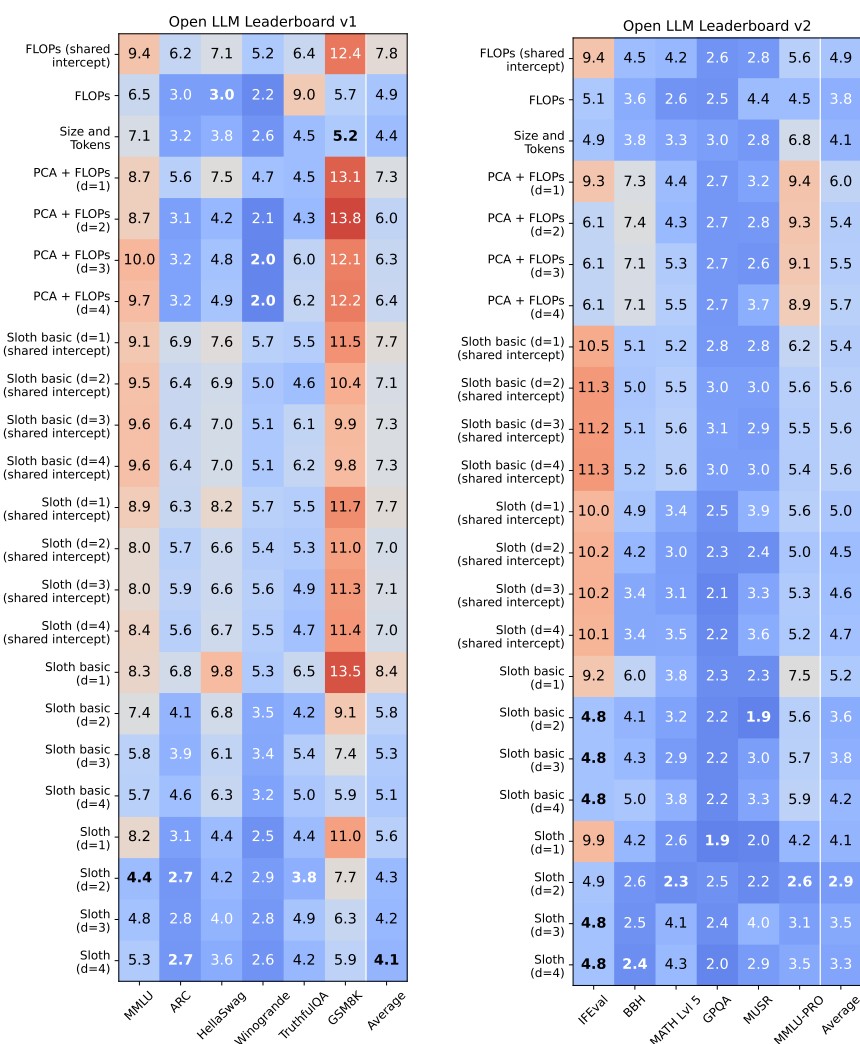

Figure 11: The figure shows the average (across LLM families) mean-absolute-error (MAE) (within a family) for different methods. This is a complete version of Figure 1, in which we include `Sloth` versions with shared intercept.

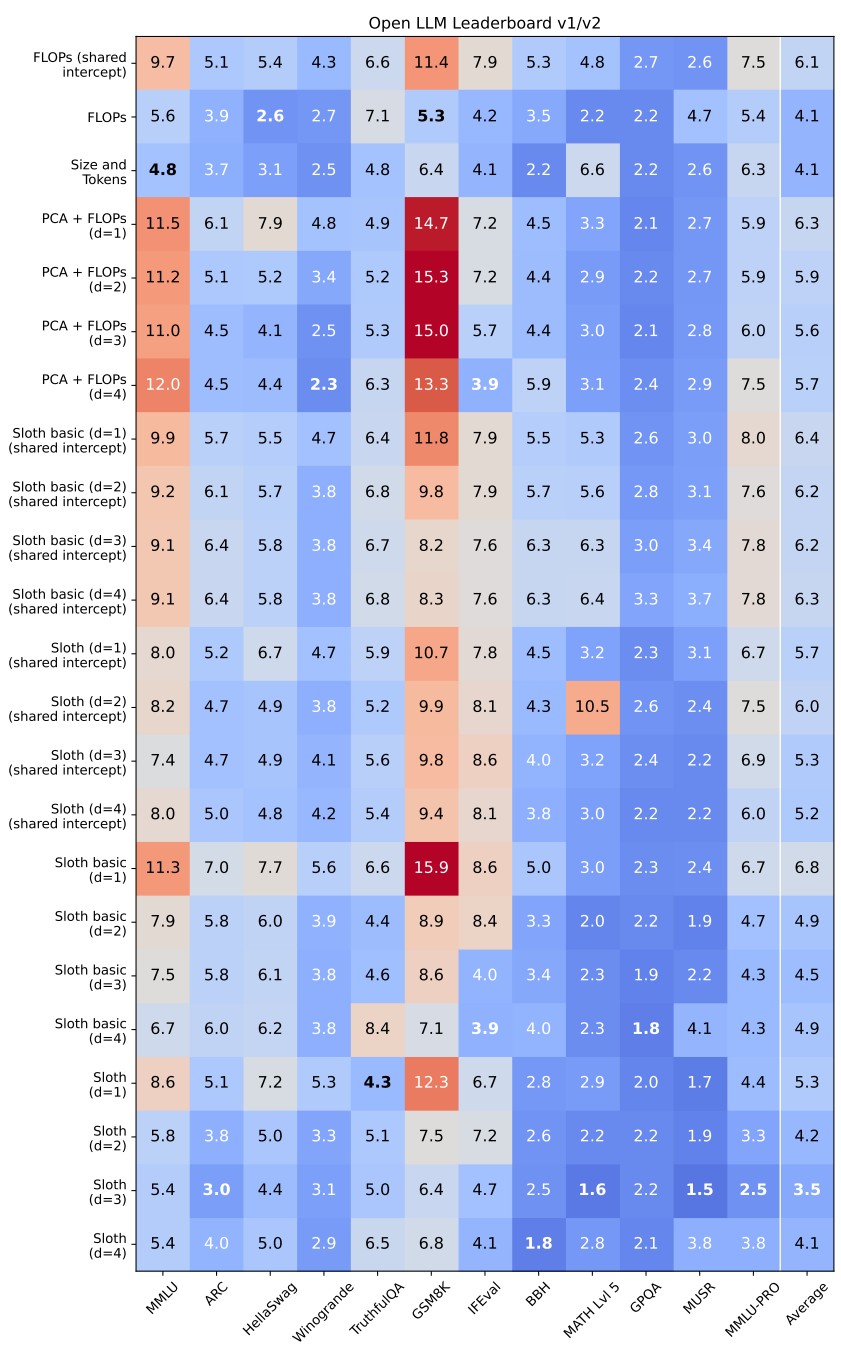

Figure 12: The figure shows the average (across LLM families) mean-absolute-error (MAE) (within a family) for different methods when fitting only one scaling law for both leaderboards.

## H.2.2 Family-specific prediction losses

Open LLM Leaderboard v1 (Average error across benchmarks)

| | bloom | codegen-nl | codellama | deepseek-coder-base | pythia | falcon | gemma | gpt-j-neo-neox | internlm2 | meta-llama-3 | mpt | olmo | opt | qwen2 | rwkv-4-pile | starcoder2 | stablelm-base-alpha | xglm | yi-1.5 |
|---|---|---|---|---|---|---|---|---|---|---|---|---|---|---|---|---|---|---|---|
| FLOPs (shared intercept) | 5.3 | 2.6 | 12.2 | 13.1 | 2.6 | 9.1 | 8.0 | 2.4 | 12.3 | 10.5 | 4.0 | 7.3 | 3.9 | 9.8 | 2.4 | 13.8 | 12.7 | 5.1 | 11.0 |
| FLOPs | 4.4 | 2.9 | 6.4 | 6.9 | 5.1 | 9.1 | 9.1 | 7.8 | 3.9 | 2.9 | 4.6 | 0.8 | 6.9 | 8.7 | 4.0 | 3.2 | 1.3 | 3.0 | 2.3 |
| Size and Tokens | 2.8 | 2.8 | 5.3 | 6.7 | 7.7 | 6.2 | 6.9 | 5.6 | 2.4 | 3.3 | 4.9 | 4.6 | 6.5 | 5.2 | 1.3 | 6.9 | 1.2 | 2.0 | 1.8 |
| PCA + FLOPs (d=1) | 5.3 | 6.6 | 8.0 | 4.7 | 7.9 | 10.2 | 10.8 | 8.8 | 5.3 | 6.3 | 8.5 | 7.9 | 9.4 | 12.7 | 6.6 | 7.3 | 2.7 | 4.8 | 5.3 |
| PCA + FLOPs (d=2) | 5.4 | 2.2 | 7.1 | 3.1 | 6.7 | 9.2 | 10.2 | 8.5 | 4.4 | 6.2 | 3.6 | 4.1 | 9.4 | 13.6 | 6.4 | 4.5 | 2.0 | 4.7 | 3.5 |
| PCA + FLOPs (d=3) | 5.5 | 1.7 | 7.0 | 3.4 | 7.5 | 11.2 | 10.5 | 9.5 | 3.4 | 6.4 | 4.7 | 2.9 | 10.3 | 13.9 | 6.9 | 4.5 | 2.0 | 5.3 | 3.9 |
| PCA + FLOPs (d=4) | 5.6 | 2.3 | 7.0 | 3.9 | 7.6 | 11.1 | 10.6 | 9.7 | 3.0 | 5.8 | 4.6 | 2.9 | 10.4 | 14.4 | 7.0 | 4.2 | 2.1 | 5.3 | 3.8 |
| Sloth basic (d=1) (shared intercept) | 6.6 | 2.8 | 12.5 | 13.0 | 3.0 | 8.6 | 6.4 | 2.6 | 12.2 | 9.6 | 4.1 | 8.1 | 4.0 | 8.7 | 2.6 | 14.0 | 12.5 | 4.5 | 10.5 |
| Sloth basic (d=2) (shared intercept) | 4.5 | 1.7 | 11.9 | 12.1 | 2.2 | 8.1 | 6.9 | 2.3 | 12.1 | 9.8 | 3.2 | 9.2 | 3.5 | 5.6 | 2.1 | 13.0 | 12.9 | 4.9 | 9.4 |
| Sloth basic (d=3) (shared intercept) | 4.9 | 1.7 | 13.0 | 12.5 | 1.8 | 8.6 | 6.8 | 2.3 | 11.9 | 13.6 | 2.7 | 9.2 | 3.2 | 5.3 | 2.1 | 13.1 | 12.8 | 4.9 | 9.2 |
| Sloth basic (d=4) (shared intercept) | 5.0 | 1.7 | 13.0 | 12.5 | 1.8 | 8.6 | 6.8 | 2.3 | 11.9 | 13.6 | 2.6 | 9.2 | 3.2 | 5.3 | 2.1 | 13.1 | 12.8 | 4.9 | 9.2 |
| Sloth (d=1) (shared intercept) | 2.5 | 4.6 | 15.3 | 15.1 | 2.3 | 9.8 | 5.7 | 3.0 | 6.4 | 7.4 | 5.1 | 10.9 | 4.8 | 9.7 | 2.3 | 14.7 | 14.2 | 5.6 | 6.6 |
| Sloth (d=2) (shared intercept) | 3.6 | 3.1 | 14.8 | 14.7 | 1.8 | 6.2 | 5.9 | 2.4 | 10.5 | 5.3 | 2.2 | 9.1 | 2.3 | 6.9 | 1.7 | 16.4 | 15.5 | 4.8 | 5.7 |
| Sloth (d=3) (shared intercept) | 3.1 | 2.6 | 14.4 | 14.9 | 1.5 | 6.5 | 6.3 | 2.3 | 11.7 | 5.8 | 2.1 | 10.0 | 2.2 | 6.9 | 1.5 | 17.0 | 14.7 | 4.7 | 6.1 |
| Sloth (d=4) (shared intercept) | 3.3 | 2.7 | 14.9 | 14.9 | 1.5 | 6.7 | 6.1 | 2.1 | 11.7 | 5.3 | 2.5 | 10.4 | 2.2 | 5.5 | 1.7 | 16.8 | 15.3 | 4.9 | 5.6 |
| Sloth basic (d=1) | 5.8 | 3.1 | 10.4 | 12.3 | 7.2 | 5.4 | 12.6 | 8.5 | 8.3 | 5.6 | 3.9 | 3.9 | 5.9 | 32.9 | 6.1 | 14.9 | 3.5 | 4.6 | 4.1 |
| Sloth basic (d=2) | 3.5 | 3.2 | 8.3 | 6.5 | 7.3 | 13.2 | 7.4 | 7.6 | 4.4 | 5.2 | 6.7 | 2.0 | 5.7 | 10.1 | 6.1 | 5.1 | 1.6 | 3.6 | 3.3 |
| Sloth basic (d=3) | 3.2 | 2.6 | 6.5 | 9.1 | 6.0 | 10.2 | 8.9 | 6.0 | 5.0 | 6.8 | 5.9 | 1.9 | 3.0 | 10.4 | 4.5 | 4.0 | 1.5 | 2.9 | 3.0 |
| Sloth basic (d=4) | 4.4 | 2.3 | 6.8 | 6.6 | 6.1 | 9.0 | 8.9 | 6.6 | 4.8 | 5.0 | 5.0 | 1.9 | 3.4 | 9.9 | 5.1 | 3.9 | 2.0 | 3.0 | 2.7 |
| Sloth (d=1) | 3.2 | 4.2 | 8.8 | 9.6 | 7.2 | 6.4 | 6.9 | 2.1 | 3.9 | 4.8 | 3.6 | 10.5 | 2.5 | 12.5 | 2.3 | 11.8 | 1.1 | 2.0 | 2.9 |
| Sloth (d=2) | 2.4 | 4.2 | 5.8 | 10.2 | 7.7 | 8.3 | 6.6 | 1.9 | 3.0 | 3.1 | 6.5 | 2.7 | 2.0 | 3.7 | 1.9 | 5.5 | 1.6 | 2.2 | 2.2 |
| Sloth (d=3) | 2.8 | 2.1 | 5.3 | 9.5 | 7.6 | 6.8 | 6.6 | 5.4 | 3.6 | 4.6 | 4.0 | 2.6 | 2.0 | 4.7 | 1.3 | 7.3 | 1.2 | 1.8 | 1.7 |
| Sloth (d=4) | 2.2 | 3.5 | 6.0 | 10.2 | 7.4 | 6.5 | 6.1 | 1.3 | 3.5 | 3.7 | 5.1 | 2.6 | 2.0 | 5.8 | 1.3 | 5.1 | 0.9 | 1.7 | 2.1 |

Figure 13: The figure shows the average (across benchmarks) mean-absolute-error (MAE) for each family considering only Open LLM Leaderboard v1.

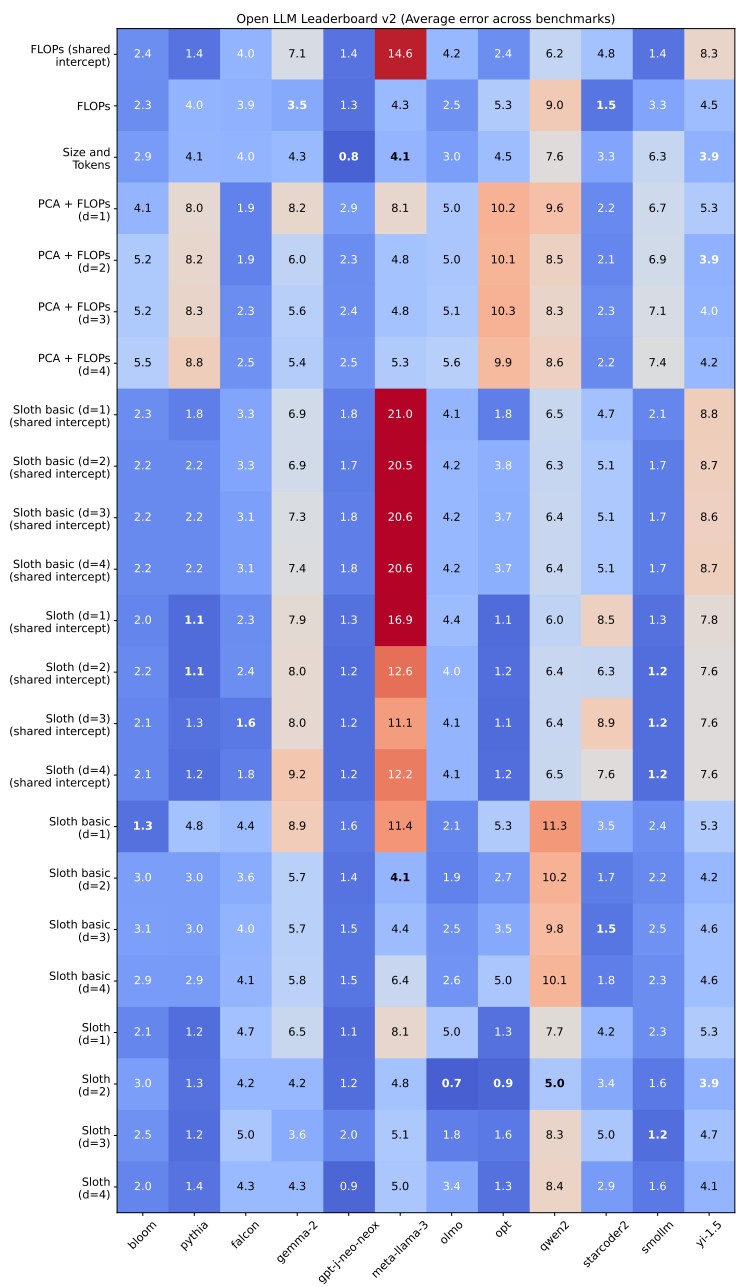

Figure 14: The figure shows the average (across benchmarks) mean-absolute-error (MAE) for each family considering only Open LLM Leaderboard v2.

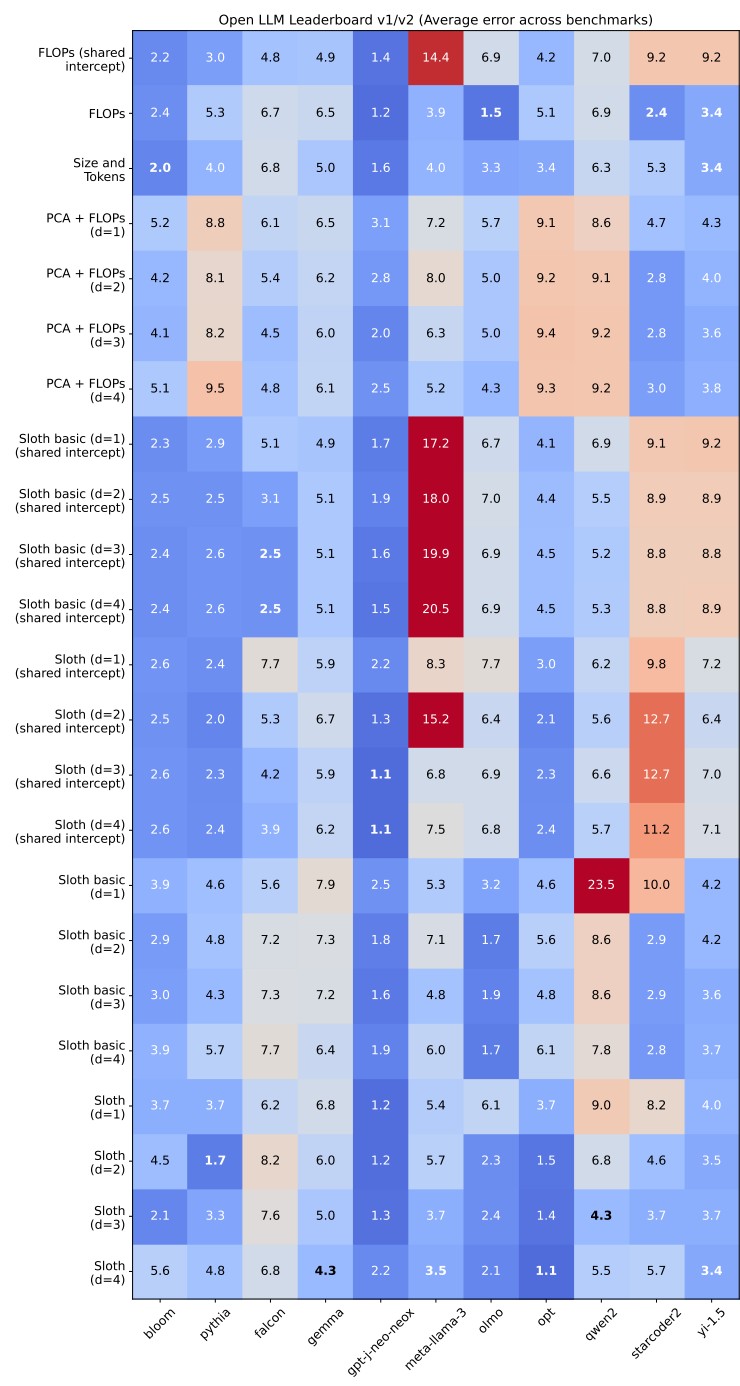

Figure 15: The figure shows the average (across benchmarks) mean-absolute-error (MAE) for each family considering Open LLM Leaderboard v1/v2.

### H.3.1 Average prediction loss across models

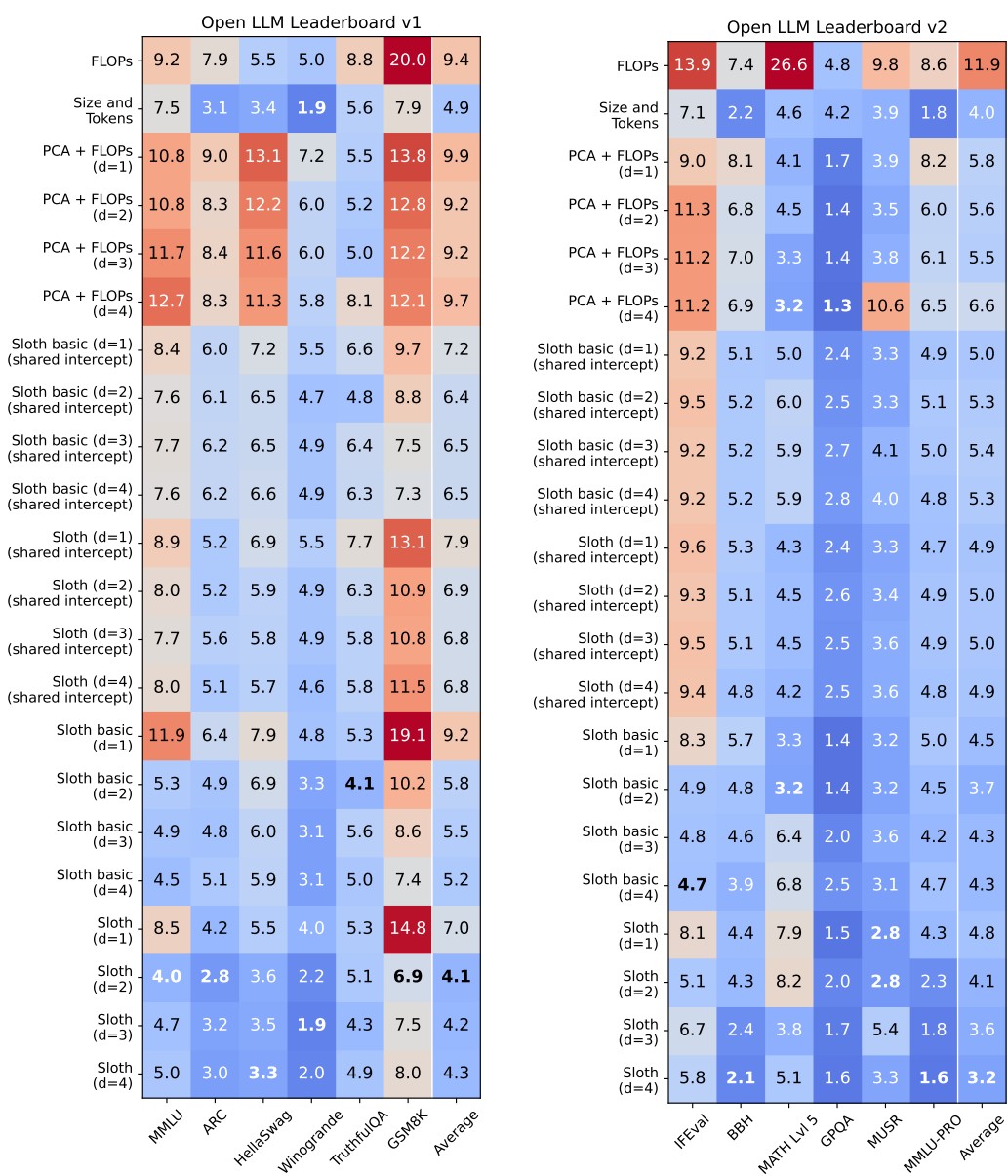

Figure 16: The figure shows the average (across LLM families) mean-absolute-error (MAE) (within a family) for different methods.

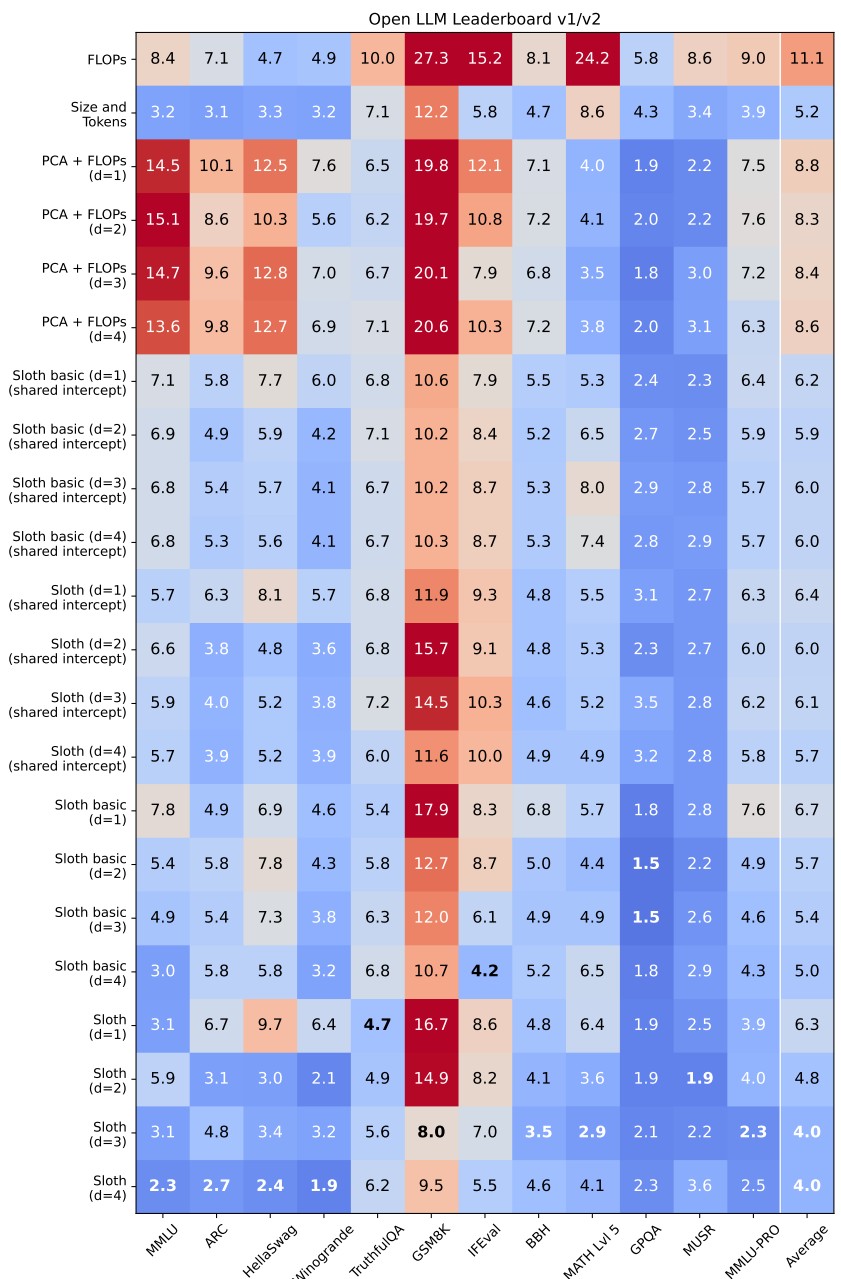

Figure 17: The figure shows the average (across LLM families) mean-absolute-error (MAE) (within a family) for different methods using the intersection of both leaderboards.

## H.3.2 Family-specific prediction losses

Open LLM Leaderboard v1 (Average error across benchmarks)

| | bloom | codellama | deepseek-coder-base | falcon | gpt-j-neo-neox | llama-2 | open_llama_ | opt | pythia | qwen1.5 | qwen2 | rwkv-4-pile | starcoder2 | xglm | yi-1.5 |
|---|---|---|---|---|---|---|---|---|---|---|---|---|---|---|---|
| FLOPs | 6.5 | 13.3 | 10.1 | 20.3 | 2.9 | 8.8 | 4.5 | 11.9 | 8.9 | 10.6 | 18.5 | 7.9 | 7.9 | 2.1 | 6.5 |
| Size and Tokens | 5.2 | 6.9 | 11.9 | 7.2 | 2.0 | 4.2 | 3.5 | 5.7 | 3.1 | 9.0 | 6.0 | **1.4** | **3.8** | **1.3** | **1.8** |
| PCA + FLOPs (d=1) | 5.2 | 9.0 | 6.4 | 17.2 | 5.5 | 4.4 | 9.3 | 6.9 | 4.7 | 7.2 | 46.1 | 4.9 | 8.4 | 2.3 | 10.9 |
| PCA + FLOPs (d=2) | 4.5 | 9.3 | 5.1 | 18.5 | 4.4 | 5.5 | 4.0 | 7.0 | 4.3 | **6.9** | 45.4 | 3.5 | 8.6 | 2.1 | 9.0 |
| PCA + FLOPs (d=3) | 4.7 | 9.2 | 4.8 | 20.5 | 3.0 | 5.1 | 4.5 | 6.4 | 4.3 | 9.0 | 43.8 | 2.4 | 8.4 | 2.5 | 8.8 |
| PCA + FLOPs (d=4) | 5.0 | 11.9 | 4.8 | 18.8 | 2.9 | 8.7 | 4.4 | 6.5 | 4.6 | 9.1 | 46.0 | 2.7 | 8.3 | 2.5 | 9.2 |
| Sloth basic (d=1) (shared intercept) | 6.2 | 10.9 | 14.5 | 11.5 | 3.6 | 5.6 | 6.1 | 5.5 | 3.4 | 8.1 | 5.4 | 3.7 | 13.2 | 3.3 | 7.4 |
| Sloth basic (d=2) (shared intercept) | 4.3 | 11.1 | 13.5 | 12.5 | 2.3 | 4.7 | 4.2 | 3.0 | 1.9 | 7.8 | **3.9** | 2.2 | 14.0 | 4.3 | 6.5 |
| Sloth basic (d=3) (shared intercept) | 4.8 | 11.6 | 14.3 | 13.4 | 2.2 | 4.4 | 4.3 | 3.0 | **1.7** | 7.4 | 5.3 | 2.1 | 13.7 | 4.1 | 5.7 |
| Sloth basic (d=4) (shared intercept) | 4.9 | 11.6 | 14.3 | 12.9 | 2.2 | 4.4 | 4.3 | 3.0 | **1.7** | 7.4 | 4.7 | 2.1 | 13.8 | 4.1 | 5.8 |
| Sloth (d=1) (shared intercept) | 4.4 | 12.2 | 19.3 | 12.4 | 3.3 | 7.6 | 7.3 | 6.0 | 2.4 | 9.1 | 8.4 | 2.8 | 12.5 | 4.2 | 6.1 |
| Sloth (d=2) (shared intercept) | 3.5 | 12.2 | 15.5 | 12.2 | 2.6 | 4.1 | 4.8 | 2.9 | 2.4 | 7.3 | 6.8 | 2.2 | 15.6 | 4.7 | 6.3 |
| Sloth (d=3) (shared intercept) | 3.7 | 12.1 | 16.0 | 12.1 | 2.5 | 4.3 | 4.2 | 2.9 | **1.7** | 7.4 | 6.6 | 1.7 | 15.2 | 4.5 | 6.5 |
| Sloth (d=4) (shared intercept) | 4.2 | 12.2 | 16.2 | 12.2 | 2.2 | 4.3 | 4.1 | 2.6 | **1.7** | 7.3 | 6.7 | 1.8 | 15.4 | 4.6 | 6.6 |
| Sloth basic (d=1) | 4.5 | 12.9 | 8.4 | 8.5 | 4.2 | 5.6 | 5.1 | 5.1 | 6.3 | 21.2 | 23.9 | 5.7 | 17.6 | 3.6 | 5.6 |
| Sloth basic (d=2) | 3.9 | 9.8 | 6.0 | 7.0 | 3.5 | 6.6 | 4.3 | 3.8 | 7.7 | 10.0 | 4.5 | 5.4 | 7.3 | 2.5 | 4.5 |
| Sloth basic (d=3) | 3.8 | 8.0 | **4.1** | 9.3 | 3.1 | 5.7 | 4.3 | 3.2 | 6.6 | 10.3 | 6.7 | 4.5 | 6.8 | 2.4 | 3.3 |
| Sloth basic (d=4) | 3.6 | 8.2 | 4.9 | 5.7 | 2.6 | 4.9 | 4.0 | 3.8 | 6.2 | 10.2 | 5.8 | 4.6 | 7.0 | 2.5 | 3.5 |
| Sloth (d=1) | 3.7 | 5.9 | 8.9 | 9.5 | 2.3 | 7.5 | 5.8 | 4.0 | 7.7 | 20.8 | 6.2 | 2.7 | 14.9 | 2.8 | 3.1 |
| Sloth (d=2) | 2.8 | **4.7** | 5.9 | 5.2 | 2.8 | 6.6 | **3.1** | 2.9 | 3.0 | 7.9 | 5.9 | 2.8 | 4.2 | 1.8 | 2.1 |
| Sloth (d=3) | 2.6 | 6.5 | 11.4 | 3.9 | 2.2 | **2.8** | 4.3 | 2.7 | 3.2 | 8.3 | 4.8 | 2.3 | 3.9 | 1.4 | 2.6 |
| Sloth (d=4) | **2.5** | 7.0 | 11.5 | **3.6** | **1.5** | 4.1 | 3.7 | **2.2** | 2.9 | 8.1 | 5.9 | 2.5 | 5.0 | 1.4 | 3.2 |

Figure 18: The figure shows the average (across benchmarks) mean-absolute-error (MAE) for each family considering only Open LLM Leaderboard v1.

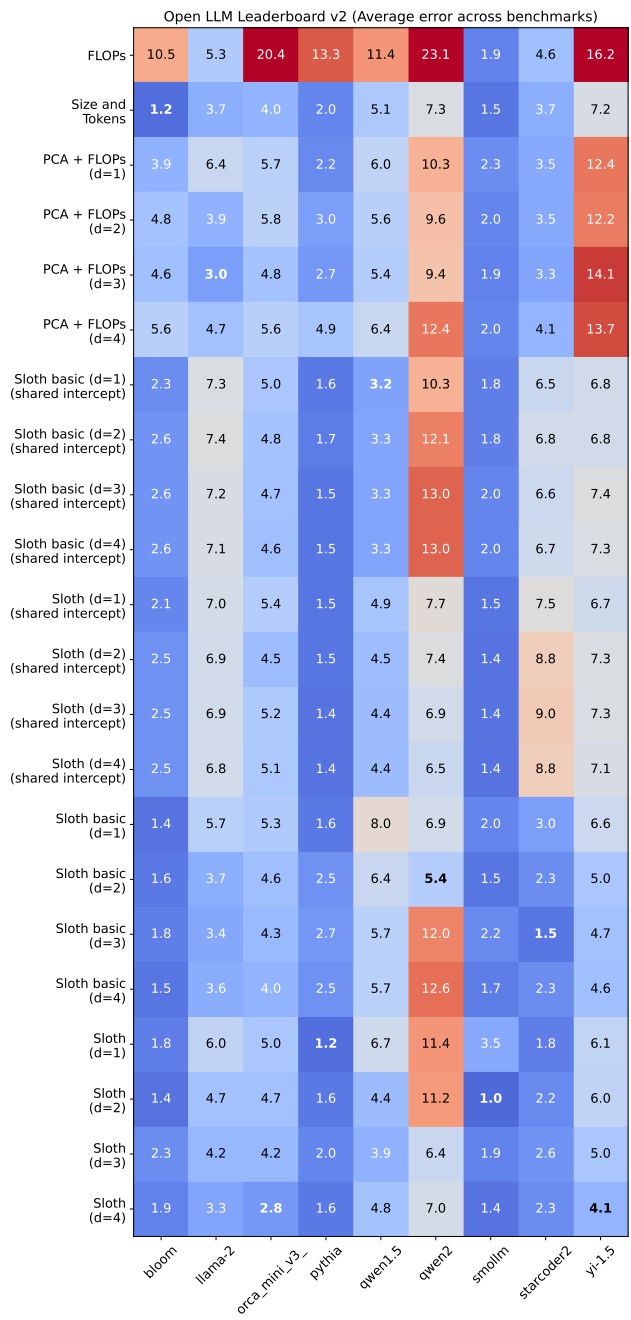

Figure 19: The figure shows the average (across benchmarks) mean-absolute-error (MAE) for each family considering only Open LLM Leaderboard v2.

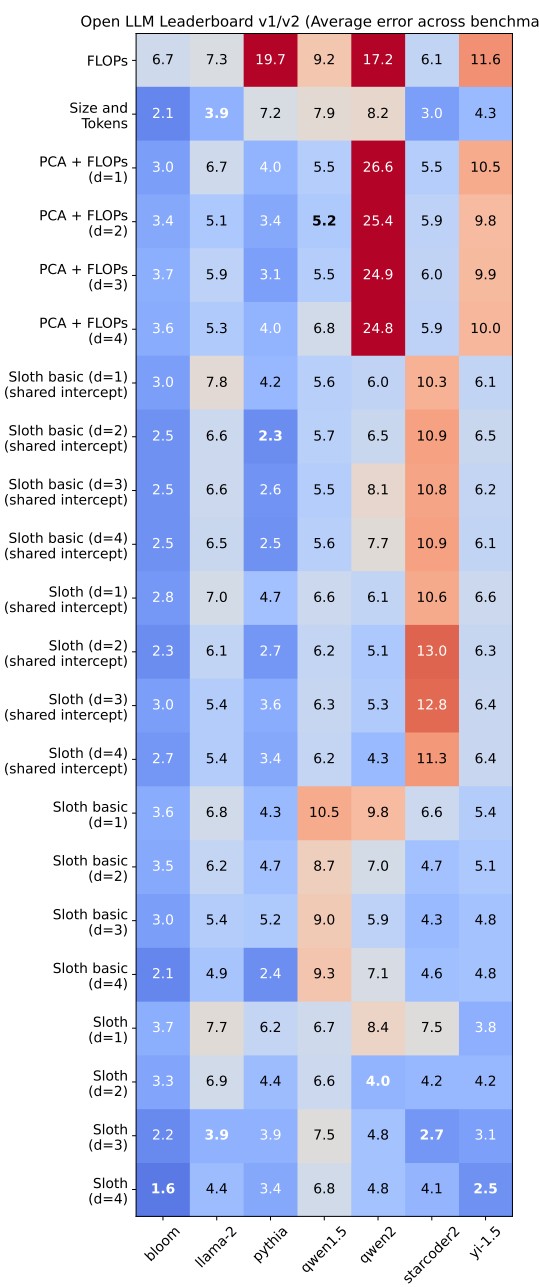

Figure 20: The figure shows the average (across benchmarks) mean-absolute-error (MAE) for each family considering only Open LLM Leaderboard v1/v2.

# I  Extra interpretability results

## I.1  Results for $d = 2$

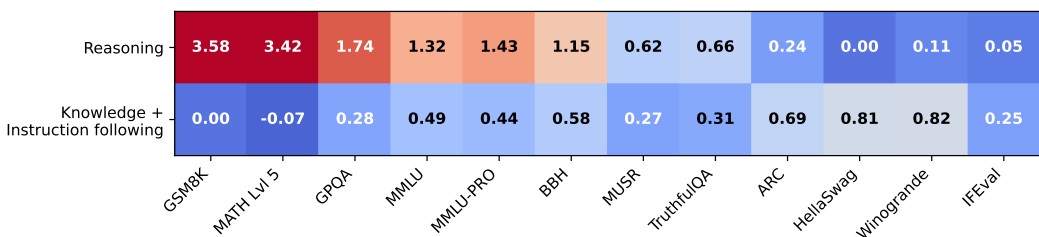

Figure 21: Needed skills for each benchmark. In this figure, we report the estimated loadings $\Lambda$ and, based on their values, we give them appropriate names.

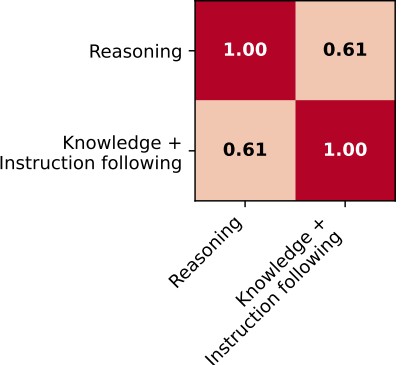

Figure 22: Needed skills for each benchmark. In this figure, we report the estimated loadings $\Lambda$ and, based on their values, we give them appropriate names.

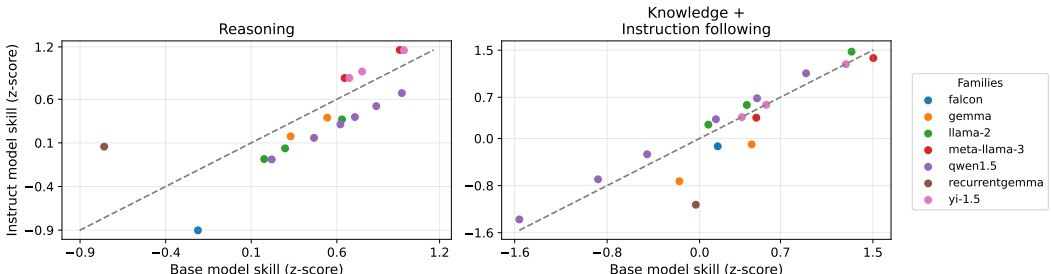

Figure 23: Gains from instruction tuning for different families on three latent skills. Major findings include a large and positive impact on instruction following and a negative impact on mathematical reasoning.

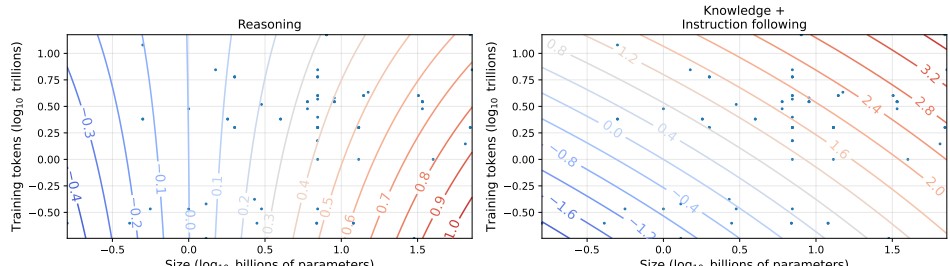

Figure 24: Level curves in producing different latent abilities from parameter count and training tokens.

## I.2 Results for $d = 3$

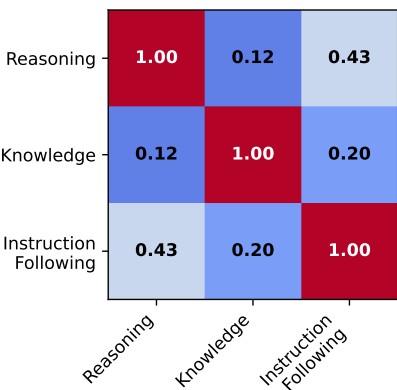

Figure 25: Needed skills for each benchmark. In this figure, we report the estimated loadings $\Lambda$ and, based on their values, we give them appropriate names.

## I.3 Results for $d = 4$

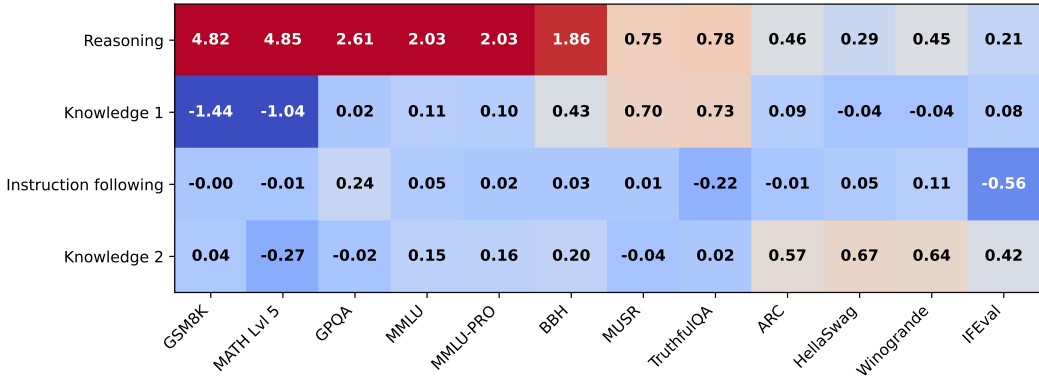

Figure 26: Needed skills for each benchmark. In this figure, we report the estimated loadings $\Lambda$ and, based on their values, we give them appropriate names.

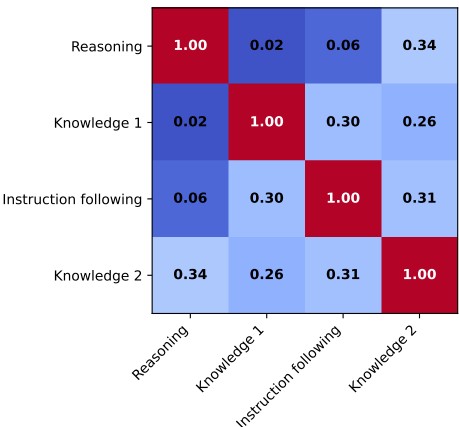

Figure 27: Needed skills for each benchmark. In this figure, we report the estimated loadings $\Lambda$ and, based on their values, we give them appropriate names.

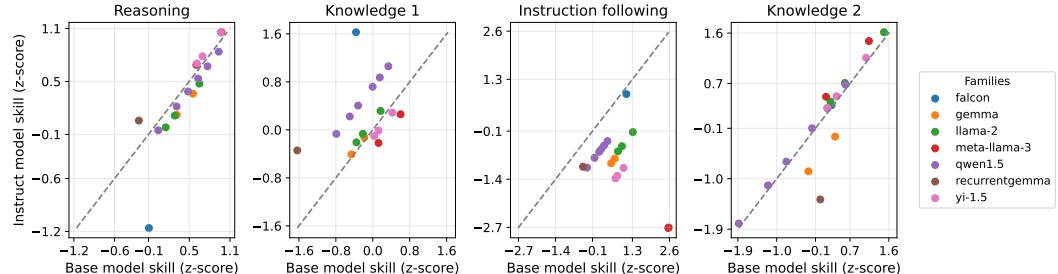

Figure 28: Gains from instruction tuning for different families on three latent skills. Major findings include a large and positive impact on instruction following and a negative impact on mathematical reasoning.

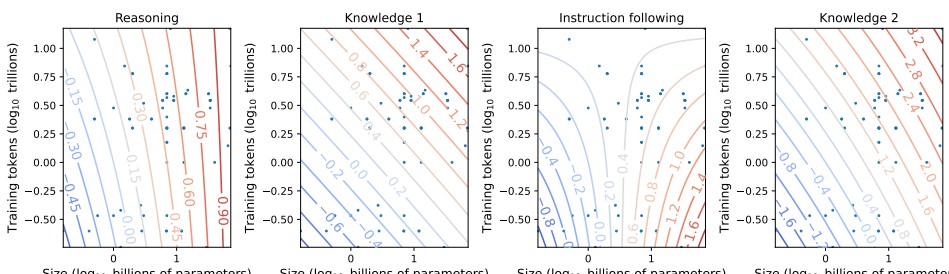

Figure 29: Level curves in producing different latent abilities from parameter count and training tokens.

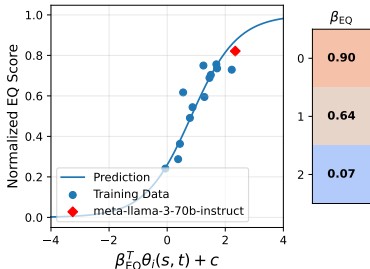

Figure 30: Predicting model performance in complex downstream tasks like emotional intelligence ("EQ") for LLaMa 3 70B (base/instruct). In the first step, we fit `Sloth` without including LLaMa 3 70B (base/instruct) in the training set. In the second step, we fit a regression model connecting skills and downstream performance. Finally, we predict LLaMa 3 70B (instruct) performance from their predicted `Sloth` skills.

## J Extra downstream task plots

### J.1 Emotional Intelligence

### J.2 Agentic Capabilities

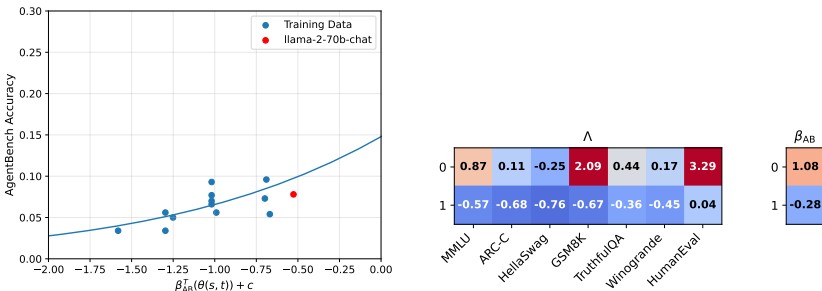

Figure 31: Predicting Agentic Capabilities of Llama-2-70B-chat.

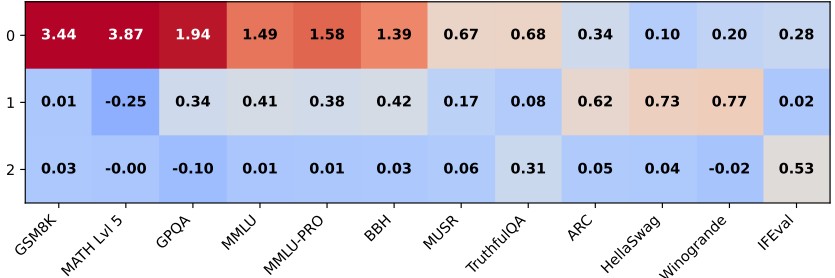

Figure 32: Loadings for downstream prediction tasks.

## K Insights from the different link functions

In this section, we visually compare `Sloth` considering trainable and logistic link function $\sigma$, Owen [2024]'s model ("FLOPs (shared intercept)") and our adaptation of Ruan et al. [2024]'s observational scaling law ("PCA + FLOPs") described in Appendix E. For this experiment, we study the two Open LLM Leaderboards separately and consider LLaMa-3 and Yi-1.5 families as the test families; we make this choice because both families are popular ones and the training set size is the same for all models in each family, making comparison between models possible (in the x-axis, we use model size). For LLaMA-3, we just include one model from that family in the training set and do not train a family-specific slope for PCA+FLOPs. For Yi-1.5, we include two models in the training

set and train a family-specific slope for PCA+FLOPs. In summary, we see that: (i) training the link function can produce a much more flexible scaling law that can better predict performance saturation (*e.g.*, the performance of Yi-1.5 in ARC, HellaSwag *etc.*), (ii) training no family-specific parameters at all ("FLOPs (shared intercept)") usually produce poor prediction results, and (iii) PCA+FLOPs often produces flatter curves that underestimate the performance of bigger models, *e.g.*, see Yi-1.5 in TruthfulQA, GSM8k, and MMLU.

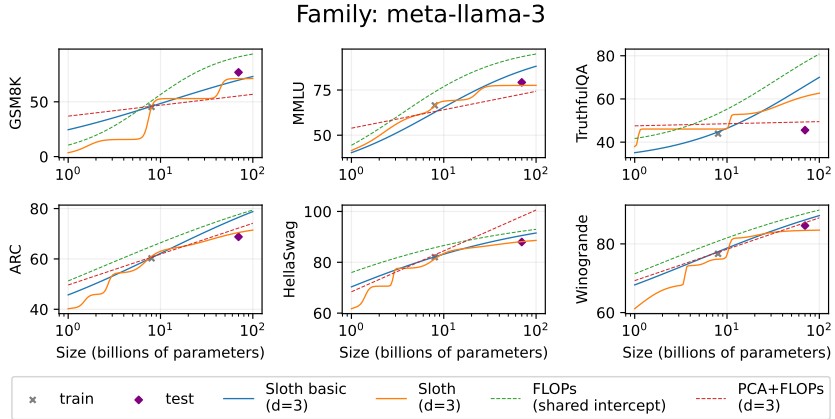

Figure 33: Prediction curves for different methods considering Open LLM Leaderboard 1 benchmark and the LLaMa-3 as the test family.

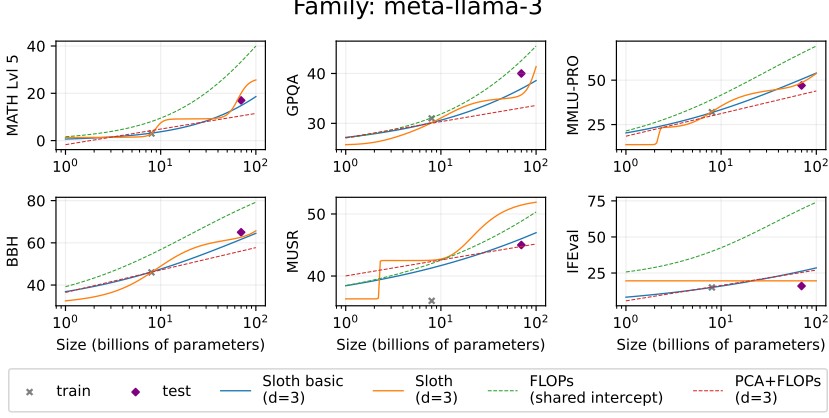

Figure 34: Prediction curves for different methods considering Open LLM Leaderboard 2 benchmark and the LLaMa-3 as the test family.

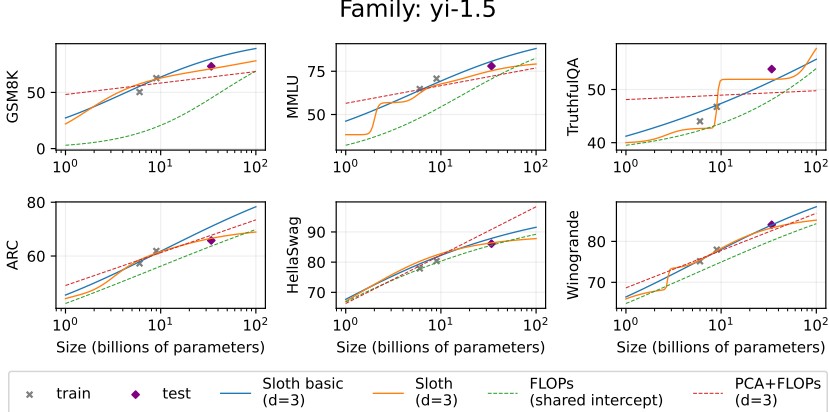

Figure 35: Prediction curves for different methods considering Open LLM Leaderboard 1 benchmark and the Yi-1.5 as the test family.

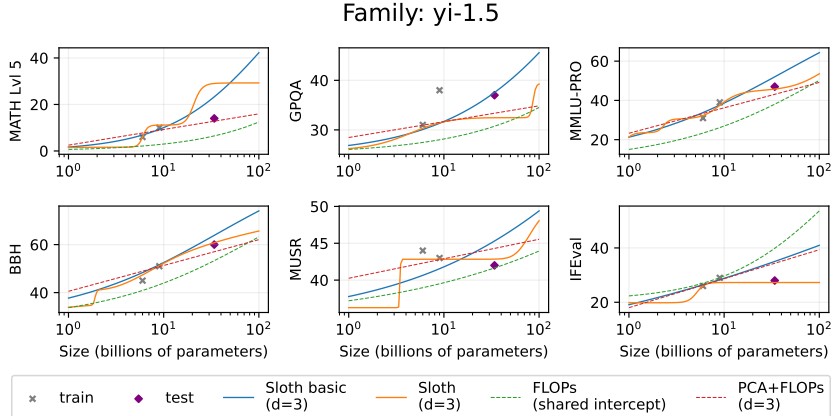

Figure 36: Prediction curves for different methods considering Open LLM Leaderboard 2 benchmark and the Yi-1.5 as the test family.

## L   Comparing against Ruan et al. [2024] in their observational scaling law setting

In this section, we compare `Sloth` with Ruan et al. [2024]'s observational scaling law; that is, we extract abstract skills using a set of benchmark scores and then use those skills to predict the performance of models of interest in a target downstream task. For this experiment, we use the same data and tasks explored in Section 4.4. For our method, we fit `Sloth` using benchmark data from all models, including performance data of LLaMa-3-70B models, and extract the skills of each model. For Ruan et al. [2024]'s method, we fit PCA on the benchmark data to extract the skills. For both methods, we set $d = 3$ and then fit a regression model with a logistic link to predict downstream performance from skills. Figures 37 and 38 present the prediction results for both methods and Figures 39 and 40 give the loading of both approaches. In both plots, out-sample prediction has a similar prediction error. At the same time, the in-sample fit is better for `Sloth` in the coding task and for Ruan et al. [2024]'s observational scaling law in the emotional intelligence task. Regarding the loading, it is possible to draw some similarities, *e.g.*, the presence of instruction following skill, but there is no one-to-one correspondence between skills.

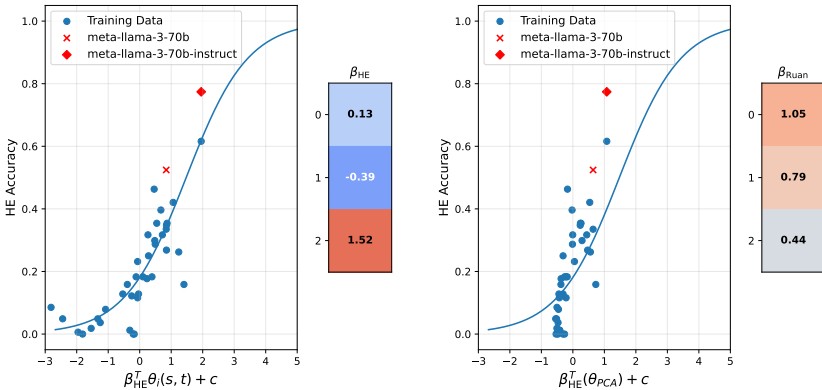

Figure 37: Predicting code completion of LLaMa 3 70B (base/instruct) with `Sloth` vs Obs. Scaling Law [Ruan et al., 2024].

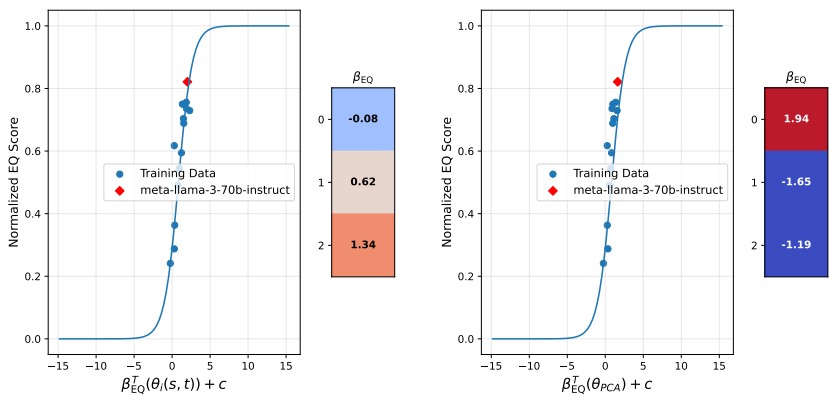

Figure 38: Predicting emotional intelligence of LLaMa 3 70B (base/instruct) with `Sloth` vs Obs. Scaling Law [Ruan et al., 2024].

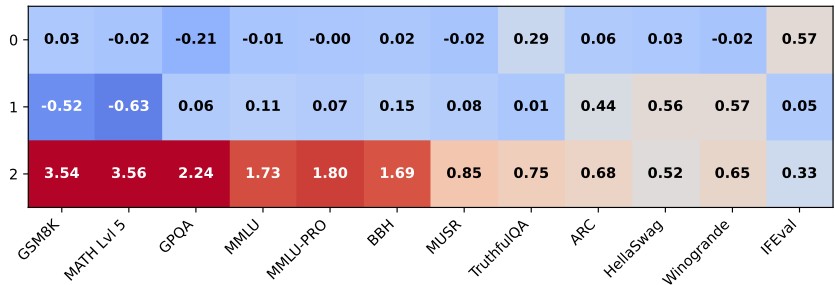

Figure 39: Loadings for downstream prediction tasks (`Sloth`).

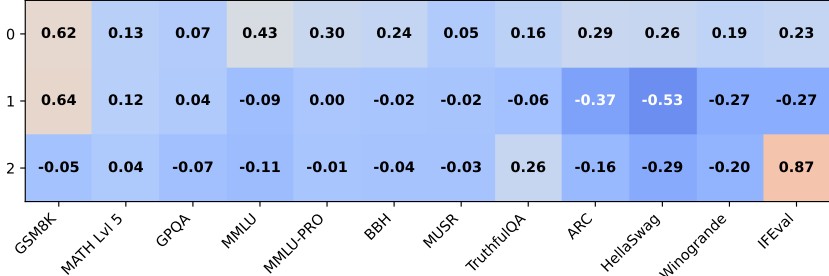

Figure 40: Loadings for downstream prediction tasks (Obs. Scaling Law [Ruan et al., 2024]).

