# OpenReview forum: "Sloth: scaling laws for LLM skills to predict multi-benchmark performance across families"
_NeurIPS.cc/2025/Conference — NeurIPS 2025 poster_

### Official Review · Reviewer_vnid · 2025-07-01

**Clarity:** 3
**Significance:** 4
**Originality:** 3
**Rating:** 5
**Confidence:** 4

**Summary:**

The core contribution of this paper is the proposal of Skills Scaling Laws (SSLaws), or "Sloth". This new class of scaling laws aims to address the limitations of existing scaling laws, which often struggle to generalize across different LLM families due to variations in training configurations and data processing. Sloth posits that LLM performance on various benchmarks is driven by a set of low-dimensional, interpretable latent skills, such as reasoning, knowledge, and instruction following. Instead of directly modeling benchmark scores as a function of computational resources (like model size and training tokens), Sloth models how these underlying latent skills scale with compute.

A key innovation of Sloth is its ability to leverage publicly available benchmark data across multiple LLM families. It assumes that while the efficiency of converting computational resources into skill levels may vary across families, the fundamental influence of resources on skills is similar. This allows Sloth to "borrow strength" across benchmarks and families, leading to more accurate and interpretable predictions. Furthermore, Sloth provides theoretical results on parameter identification and demonstrates practical applications, including predicting performance in complex downstream tasks, understanding behavior with scaled test-time compute, and deriving optimal-scaling rules for skills given a compute budget.

**Questions:**

Q1: why is theta considered to be latent skills or `reasoning'? What's your clue and what's the justification?
Q2: If I understand correctly, your alpha is family-dependent variable. How do you model the family with only the smallest model given, as described in L223-236?

Q3: l252, Which figure shows the results of the two observed models? Your link directs to a bunch of figures.

**Ethical Concerns:**

["NO or VERY MINOR ethics concerns only"]

**Final Justification:**

I think this paper is of good quality and the author's response to my question looks good to me. Thus, I retain my score.

**Limitations:**

Yes

**Quality:**

4

**Strengths And Weaknesses:**

Strength:
This paper has several clear strengths.
1. Address important limitations of previous work: I appreciate the authors' taking Section 2 to compare against previous studies and better position the paper's contributions. Previous work either ignores the family-dependent feature of the scaling law or has the model first then predict the performance on non-evaluated benchmarks. This paper addresses these limitations by encoding the correlation structure between benchmarks.

2. Practical experiments and setups: The Authors evaluated their methods in a practical setting where only one or two models with the smallest sizes are trained, and predict the performance of the large models.

3. Improved predictive accuracy and generalizability: Experimental results consistently show that Sloth performs competitively, often achieving similar or lower mean absolute errors than baselines, even for recent and diverse LLM families like LLaMa 3, Qwen 2, and Yi 1.5.

Weakness:

I found no critical weaknesses.

---

> ### Author Rebuttal · Authors · 2025-07-30
>
> Thank you for taking the time to review our paper. We have responded to your comments and questions below. If you have any further feedback or concerns, please feel free to let us know.
>
> -----------
>
> ### **Questions**
>
> **Q1: Why is theta considered to be latent skills or “reasoning”? What is the justification?**
>
> The $\theta$ parameters are low-dimensional embeddings for each LLM, capturing what the models are capable of. We demonstrate that model performance can be predicted using just these tiny embeddings. Referring to them as “skills” or “abilities” follows conventions in psychometrics, which studies latent traits like human intelligence using models like ours (e.g. factor analysis, item response theory, etc).
>
> ---
>
> **Q2: If I understand correctly, your alpha is a family-dependent variable. How do you model the family with only the smallest model given, as described in L223-236?**
>
> When fitting Sloth (see the optimization problem in Appendix D), we only need to observe a single LLM from the target family to estimate its family-specific parameter ($\alpha_i$). This is analogous to fitting a linear regression with group-specific intercepts, often referred to as fixed effects in statistics. Naturally, for some families (not necessarily the target of interest), we do need to observe models of different model/training set sizes to estimate the shared slope parameters ($\beta_k$’s). We give you a brief intro to linear regression with group-specific intercepts below, so it can solidify the intuition. If anything remains unclear, please let us know.
>
> *Linear Regression with Group-Specific Intercepts (Fixed Effects):* In a standard linear regression, we model the outcome $y$ as
>
> $$
> y_{ij} = \beta_0 + \beta_1 x_{ij} + \epsilon_{ij}
> $$
>
> where $i$ indexes groups (e.g., model families) and $j$ indexes observations within group $i$.
>
> To account for group-specific baseline differences, we extend the model by allowing each group to have its own intercept:
>
> $$
> y_{ij} = \alpha_i + \beta_1 x_{ij} + \epsilon_{ij}
> $$
>
> where $\alpha_i$ is the intercept specific to group $i$. Here, $\beta_1$ is shared across all groups, but each group can start from a different baseline level via $\alpha_i$. The fixed effects $\alpha_i$ can be estimated for a target group by only observing one individual in that group as long for some training groups we observe more than one individual (needed to estimate $\beta_1$).
>
>
> ---
>
> **Q3: l252, Which figure shows the results of the two observed models? Your link directs to a bunch of figures.**
>
> Thank you for catching this! Figure 15 is directly analogous to Figure 1 and presents the results for the two observed models. We will clarify this in the revision.

---

> > ### Comment · Reviewer_vnid · 2025-08-05
> >
> > Thanks for your response to my question. I will retain my score.

---

### Official Review · Reviewer_mbUa · 2025-07-03

**Clarity:** 3
**Significance:** 2
**Originality:** 1
**Rating:** 3
**Confidence:** 4

**Summary:**

This paper proposes to include model family to estimate scaling laws across data and models.

**Questions:**

- How much data/benchmarks are needed to make predictions for a new model family? Is this exacerbated given the heavy tailed nature of the data? How does this relate to the diversity of benchmarks?
- How much does the predictive performance change as you add more samples? Given that
- In Figure 6, sloth overestimates the performance of pythia, can you elaborate for which scenarios the proposed model  over vs under estimates the performance?

**Ethical Concerns:**

["NO or VERY MINOR ethics concerns only"]

**Final Justification:**

As I mentioned, I appreciate the proposed framework in the paper, which leverages information from multiple benchmarks to predict model performance within families. However, as it stands “under/overestimation is sample-dependent and can vary by scenario.” This currently limits the conditions under which users can reliably apply the method.

**Limitations:**

yes

**Quality:**

3

**Strengths And Weaknesses:**

The paper is well written, and provides a through analysis across benchmarks.
The authors claim that family specific intercepts are not always needed (Figure 3), and add some worst case analysis to their results. However, these conclusions should be highlighted in the main test
The method sees little benefit from adding more samples within a model family as long as there is enough data available for a wide variety of datasets; however, outside of the model family the model can under/over estimate performance. It is unclear when it is biased towards either failure model.

---

> ### Author Rebuttal · Authors · 2025-07-30
>
> Thank you for your work on our paper. We have addressed your comments and questions below. Please let us know if you have further questions or concerns.
>
> ---
>
> ### **Questions**
>
> **Q1: How much data/benchmarks are needed to make predictions for a new model family? Is this exacerbated given the heavy-tailed nature of the data? How does this relate to the diversity of benchmarks?**
>
> - For prediction, our approach can make reliable predictions for a new model family using as little as a single benchmark (the "Size and Tokens" results in Figure 1 illustrates this), and a single model per test family (full Figure 1).
> - For skill estimation, once the Sloth weights are fitted, it’s sufficient to observe scores on $d$ (the number of skills) benchmarks, assuming they’re reasonably diverse, to estimate the family-specific parameters ($\alpha_i$) for the new family. This flexibility even accommodates missing data, though in practice, gathering benchmark results is not a significant bottleneck for the type of benchmarks we use.
> - Importantly, we do not see how heavy-tailed data can affect negatively the results.
>
>
> **Q2: How much does predictive performance change as you add more samples?**
>
> As expected for most machine learning methods, increasing the number of samples improves accuracy. In practice, however, we find that having just one model per family in the training set is often sufficient (please see Appendices J.2 vs J.3).
>
>
>
> **Q3: In Figure 6, sloth overestimates the performance of pythia. Can you elaborate on scenarios where the model over- vs. under-estimates performance?**
>
> There is no general rule: under/overestimation is sample-dependent and can vary by scenario, as is common in machine learning.
>
> ---
>
> ### **Clarifying points**
>
> - From your review, we understand that there may be some misunderstanding about our work. Our contribution is not simply proposing “to include model family to estimate scaling laws across data and models”; we propose a **scaling law for LLM skills** that **decouples model size and training tokens** and uses **family‑specific intercepts with shared skill slopes** to enable reliable extrapolation both within a family (from small to large models) and to **new families with minimal observations**. We are happy to clarify in more detail if needed. Please let us know if there is still anything unclear.
>
> - From your review ratings, you are mostly concerned with novelty and significance of our work while this is not reflected by your written review. For an explanation on novelty, please see our response to Reviewer 9m8f. For significance, you do not mention any serious limitation of our work in your review that could justify it; moreover, we have provided a diverse and practical set of five realistic applications in which our method could be applied to. Could you clarity these two points, please? Thank you.
>
> We are open for discussion and happy to clarify any misunderstanding about our work.

---

> > ### Comment · Reviewer_mbUa · 2025-08-09
> > **response**
> >
> > Thank you for your response. I appreciate the proposed framework in the paper, which leverages information from multiple benchmarks to predict model performance within families. Accordingly, I have raised my score to 4. I believe the work would benefit from additional explanation and guidelines for users, as the paper notes that “under/overestimation is sample-dependent and can vary by scenario.” This currently limits the conditions under which users can reliably apply the method.

---

### Official Review · Reviewer_9m8f · 2025-07-03

**Clarity:** 2
**Significance:** 3
**Originality:** 2
**Rating:** 4
**Confidence:** 3

**Summary:**

Due to the inherent differences across model families, existing scaling laws often fail to generalize universally across LLMs. Constructing separate scaling laws for each model family, however, incurs substantial computational cost. To address this challenge, this paper proposes Skills Scaling Laws (SSLaws, Sloth), a method that leverages publicly available benchmark data and assumes LLM performance is driven by a set of low-dimensional latent skills—such as reasoning and instruction following. By exploiting the correlation structure across benchmarks, Sloth enables more accurate and interpretable performance prediction, while alleviating the need to train multiple models per family. Empirical results demonstrate that Sloth accurately predicts LLM performance and provides valuable insights into downstream scaling behavior, test-time compute efficiency, and compute-optimal skill development.

**Questions:**

1. Are the identified latent skills strongly correlated with the choice of benchmark datasets? How does the number of these skills influence the model’s predictive performance?
2. If certain benchmarks are potentially contaminated during model training, to what extent can their evaluation results be considered trustworthy?
3. Beyond predicting model performance on benchmarks, can this scaling law offer guidance for model training or the adjustment of specific capabilities? Given that benchmark datasets may not always faithfully reflect the full spectrum of a model’s abilities, it would be valuable to understand whether the proposed method can inform capability refinement or training strategies.

**Ethical Concerns:**

["NO or VERY MINOR ethics concerns only"]

**Final Justification:**

The author’s response provided clarification regarding the similarities with Ruan’s work, which helped resolve some of my concerns. As a result, I have decided to raise my original score.

**Limitations:**

Yes

**Quality:**

3

**Strengths And Weaknesses:**

Strengths:
1. By leveraging publicly available benchmark data and the concept of latent skills, this work proposes a resource-efficient approach to model performance prediction. In addition to reducing the need for training multiple models, the method offers valuable insights into the interpretability of model capabilities.

Weaknesses:
1. The main method proposed in this paper is quite similar to that of Ruan et al., but the fundamental differences between the two are not clearly explained in the related work section or in the subsequent content.


[1] Yangjun Ruan, Chris J. Maddison, and Tatsunori Hashimoto. Observational Scaling Laws and the Predictability of Language Model Performance, July 2024.

---

> ### Author Rebuttal · Authors · 2025-07-30
>
> Thank you for your thoughtful review of our paper. We have addressed your comments and questions below. Please let us know if you have any additional questions or concerns.
>
> ----------------
>
> ### **Similarity with Ruan (2024)**
>
> We respectfully disagree in this point. Our approach differs substantially from Ruan (2024), primarily in three aspects (*all of them explained or explicit in Sections 2.2, 3, 4*):
> 1. **Different goals and setup.** While Ruan focuses on only interpreting the skills of trained LLMs (i.e., with no predictive aspect), our method is designed to predict the skills/performance of hypothetical LLMs that have not yet been trained, while retaining the interpretability aspect. *Please check our Section 2.2 for a detailed discussion.*
> 2. **Different methodology.** Our scaling law has several difference from theirs: (i) we decouple FLOPs into size and tokens, incorporating an interaction term (explained in Appendix F); (ii) we make the link function $\sigma$ trainable (if needed); (iii) we propose that slopes do not need to be family-dependent for a good performance; (iv) we propose a new fitting algorithm that allow skills being different from principal components, allowing more flexibility and better interpretability (since skills do not need to be orthogonal, for example).
> 4. **Different set of experiments.** All experiments in Sections 4.2, 4.4, 4.5, and 4.6 are not possible within Ruan's setup (and not the focus of their paper) either because they do not focus on the predictive aspect of scaling laws (4.2, 4.4, 4.5) or because they do not decouple model size and tokens (4.6) in their work. Regarding Section 4.2, we attempted to adapt Ruan's model for prediction in our experiments and find that it does not perform well (see, for example, the PCA method in Figure 1).
>
> ---
>
> ### **Questions**
>
> **Q1: Are the identified latent skills strongly correlated with the choice of benchmark datasets? How does the number of these skills influence the model’s predictive performance?** Indeed, the latent skills we identify are influenced by the set of benchmarks used; for example, if none of the benchmarks require reasoning, reasoning skills cannot be identified. However, the number of skills primarily reflects the diversity of the benchmarks and LLM abilities, not predictive accuracy per se. Regardless of the true number of skills, we can fit models with any number of skills (see Figure 1). Too few skills will underfit, while too many will overfit.
>
> ---
>
> **Q2: If certain benchmarks are potentially contaminated during model training, to what extent can their evaluation results be considered trustworthy?** Contamination is a concern for skill estimation, as it is for any application involving AI evaluation and benchmarks. Our approach does not eliminate this challenge, and it is an important caveat for interpreting results.
>
> ---
>
> **Q3: Beyond predicting model performance on benchmarks, can this scaling law offer guidance for model training or the adjustment of specific capabilities? Given that benchmark datasets may not always faithfully reflect the full spectrum of a model’s abilities, it would be valuable to understand whether the proposed method can inform capability refinement or training strategies.** Yes, our scaling law can be used to guide training strategies. As described in Section 4.6, we provide a concrete example of how the model can be used to optimize resource allocation when training LLMs, maximizing desired skills within a fixed compute budget.

---

> > ### Comment · Reviewer_9m8f · 2025-08-05
> >
> > Thank you for your detailed response and the clarification regarding the similarities with Ruan’s work. I appreciate your efforts, and I will carefully consider your points when updating my review score.

---

> ### Author Response · Authors · 2025-08-05
>
> We appreciate the reviewer’s response. To clarify the distinctions between our paper and Ruan (2024) in a more structured way, we have decided to include the following table in the next revision of our paper; thank you for your feedback. Please let us know if you have any further questions about the distinctions between the two works. We are happy to provide more details on any point that remains unclear.
>
> | Aspect                                    | **Ruan (2024)**                                                                                                | **Our paper (Sloth)**                                                                                                                         |
> | ----------------------------------------- | -------------------------------------------------------------------------------------------------------------- | --------------------------------------------------------------------------------------------------------------------------------------------- |
> | **Goals & setup**                         | Interprets the skills/capabilities of **already trained** LLMs; does **not** aim to predict untrained models.  | Designed to **predict** the skills/performance of **hypothetical (not-yet-trained)** LLMs, while retaining interpretability. *(See §2.2.)*    |
> | **Methodology — compute variables**       | Uses **FLOPs** as a single variable (does not separate size vs. tokens).                                       | **Decouples FLOPs** into size (s) and tokens (t) and includes an **interaction** term (e.g., log s × log t). *(See App. F.)*                  |
> | **Methodology — link function**           | Uses a **fixed link** between latent capability and benchmark score.                                           | Allows the **link to be trainable** (when helpful).                                                                                           |
> | **Methodology — family dependence**       | **Family-dependent slopes** (and other parameters) are assumed/needed.                                         | Shows that **slopes do not need to be family-dependent** and achieves stronger predictive performance.                                            |
> | **Methodology — skill representation**    | Skills derived from **principal components (PCs)**; orthogonality is implied by PCA; uses standard regression. | Proposes a **new fitting algorithm** where **skills are not restricted to PCs** or orthogonality, improving flexibility and interpretability. |
> | **Experiments — predictive focus**        | Not designed for **predicting** the performance of untrained models; such setups are **outside** Ruan’s focus. | Runs predictive experiments that **Ruan’s approach does not support** (focus is on the prediction of hypothetical models' skills).                |
> | **Experiments — size & token decoupling** | Does **not** decouple size and tokens; cannot test token/size trade-offs directly.                             | **Explicitly evaluates** size–token trade-offs via the decoupled, interacting specification.                                                  |

---

### Official Review · Reviewer_TtJM · 2025-07-05

**Clarity:** 3
**Significance:** 3
**Originality:** 3
**Rating:** 5
**Confidence:** 3

**Summary:**

This paper proposes Sloth, a new class of scaling laws that predicts large language model (LLM) performance across benchmarks and model families. The core idea is to represent LLM capabilities using a small set of interpretable latent skills (like reasoning and knowledge), which scale with compute. Sloth leverages leaderboard data and allows for family-specific efficiency without requiring multiple models per family. It supports flexible, optionally learned activation functions to improve prediction accuracy. The method is validated across 12 benchmarks and used to forecast downstream task performance for unseen, larger models.

**Questions:**

Please see Weakness part

**Ethical Concerns:**

["NO or VERY MINOR ethics concerns only"]

**Final Justification:**

The authors have addressed my concerns.

I keep my score.

**Limitations:**

Please see Weakness part

**Quality:**

3

**Strengths And Weaknesses:**

Strengths:

1. The paper is very well written, with clear structure and logical flow throughout.

2. The authors conduct thorough ablation studies, which help isolate the contribution of each component in the proposed method.

Weaknesses:

I did not find major weaknesses in this work.

1. I’m particularly interested in the design choices behind the proposed scaling law formulation. It would be helpful if the authors could share more motivations and potentially failed alternatives. Why is each skill slope shared across families in Equation (3.2)?

2. I’m not fully convinced by the interpretation of benchmark-skill associations in Section 4.3. In particular, I believe MMLU might be more appropriately categorized under knowledge-based skills rather than reasoning.

---

> ### Author Rebuttal · Authors · 2025-07-30
>
> Thank you for your work on our paper! We have addressed your comments and questions below. Please let us know if you have further questions or concerns.
>
> -------
>
> ### **Design choices behind our scaling law**
>
> Our scaling law is informed by the shortcomings of previous benchmark scaling models, such as those proposed by Owen (2024) and Ruan (2024). We build on these approaches by explicitly decoupling the effects of training tokens and model size, which were previously conflated through the use of FLOPs. Furthermore, we introduce an interaction term, discussed in Appendix F, which further improves the flexibility of the model. While we also explored higher-order (quadratic) terms, we found these unnecessary for capturing the relevant trends.
>
> As highlighted in Section 2 and in our experimental results, earlier models tended to be either overly flexible, making both intercept and slope family dependent, which hinders reliable estimation when only a few models are available per family, or too constrained, failing to capture differences across LLM families. In our setting, allowing slopes to vary by family would be even more challenging given that our model incorporates more covariates than previous work. By choosing to make only the intercepts family-specific while keeping the skill slope shared, we achieve a balance: Our approach enables robust extrapolation to larger models based on observations from just a few (or even a single) smaller model(s). Beyond these empirical findings, there is also a theoretical rationale for favoring family-specific intercepts with shared slopes, which we outline in the following section.
>
> **More intuition behind our formulation with family-specific intercepts (and shared slopes):** This formulation is inspired by classical models in economics (as mentioned in the text), where intercepts represent the efficiency of a firm or production unit. Similarly, we interpret the family-specific intercept as a measure of 'model family efficiency', directly affecting how input levels (such as training tokens or model size) translate into performance. This 'model family efficiency’ term could absorb things like the quality of the training data or changes in architecture.
>
> To make this concrete, consider a simplified case, assuming the benchmark performance is affected by only one skill in Eq 2.2, that $\gamma=0$, and in which we assume no interaction term of $\log s$ and $\log t$. Concretely, assume that the expected performance in that benchmark is given by
> $$
> \sigma(\alpha_i + \beta^\top x(s, t)) = \frac{e^{\alpha_i + \beta_1 \log s + \beta_2 \log t}}{1 + e^{\alpha_i + \beta_1 \log s + \beta_2 \log t}}
> = \frac{A_i s^{\beta_1} t^{\beta_2}}{1 + A_i s^{\beta_1} t^{\beta_2}},
> $$
>
> where $A_i = e^{\alpha_i}$ captures the efficiency of family $i$.
> This highlights how the impact of model size ($s$) and training tokens ($t$) on performance fundamentally depends on the family intercept $\alpha_i$: for families with higher efficiency ($\alpha_i$), increases in $s$ and $t$ yield a bigger impact on performance. This intuition first guided our choice to make only the intercept family-specific.
>
> In summary, our design choices are grounded in both empirical considerations (robustness and generalizability across families) and theoretical intuition (interpreting intercepts as efficiency), resulting in a model that is both interpretable and practically effective.
>
> ---------
>
> ### **Benchmark-skill associations**
>
> Thank you for this observation. When we refer to "Knowledge," we specifically mean “common-sense knowledge," (as explained from line 287 onwards), and when we refer to "Reasoning", it can also incorporate advanced STEM knowledge (which is the case of MMLU). To clarify, we will label "Knowledge" as “Common-sense knowledge"  and "Reasoning" as "Reasoning and advanced knowledge" in the text. Thank you!

---

> > ### Comment · Reviewer_TtJM · 2025-08-06
> > **Thanks for your response**
> >
> > I will maintain my score.

---

### Decision · Program_Chairs · 2025-09-17

**Decision:**

Accept (poster)

**Comment:**

# Summary
This paper proposes skills scaling laws (SSLaws) that trains a model to predict the performance of a model from a specified family on a specified benchmark, given the model size and the amount of training tokens. The SSLaws model is trained on a predefined set of model families and a set of benchmarks, and can generalize to untrained models associated with an unseen configuration of (model size, training token number) from one of the model families. Compared to existing scaling law models, different benchmarks in SSlaws share the same representation space defined by low-dimensional latent skills, and the effect of increasing model sizes and training tokens is reflected on the skill representations. Experiments demonstrate that SSlaws accurately predicts LLM performance and provides valuable insights into downstream scaling behavior, test-time compute efficiency, and compute-optimal skill development.

# Strengths
- The paper is well-written with a clear logical flow.
- The ablation studies and experimental results are comprehensive and show the advantages of SSlaws.
- The generalization from one or two small models per family to other model configurations is non-trivial.
- The method captures the correlation within the same model family and the shared skills, which are novel compared to existing works.

# Weaknesses
- The paper can be improved by providing more discussion and clarification about the motivations, other alternative approaches, and benchmark-skill associations.
- It is not fully clear how the approach can predict other specific capabilities of LLMs and guide training.
- A more detailed comparison to some related works is needed.
- Insufficient analysis about the main factors affecting the prediction accuracy, e.g., the number of training data.

# Reasons to Accept
- The paper proposed novel parameter-sharing strategies (latent skills, skill slopes, etc.) to improve the generalization of the scaling law prediction model.
- The proposed method is effective across different model families and benchmarks in the experiments.
- Most reviewers support the acceptance.

# Discussion Summary
- In the rebuttal, the authors provided detailed responses with additional details to answer the questions of the reviewers.
- All reviewers confirmed that they are satisfied with the responses.